# FUNCTION-SPACE PARAMETERIZATION OF NEURAL NETWORKS FOR SEQUENTIAL LEARNING

**Aidan Scannell**,[*] **Riccardo Mereu**,[*] **Paul Chang, Ella Tamir, Joni Pajarinen & Arno Solin**
Aalto University, Espoo, Finland
`{aidan.scannell,riccardo.mereu}@aalto.fi`

## ABSTRACT

Sequential learning paradigms pose challenges for gradient-based deep learning due to difficulties incorporating new data and retaining prior knowledge. While Gaussian processes elegantly tackle these problems, they struggle with scalability and handling rich inputs, such as images. To address these issues, we introduce a technique that converts neural networks from weight space to function space, through a dual parameterization. Our parameterization offers: *(i)* a way to scale function-space methods to large data sets via sparsification, *(ii)* retention of prior knowledge when access to past data is limited, and *(iii)* a mechanism to incorporate new data without retraining. Our experiments demonstrate that we can retain knowledge in continual learning and incorporate new data efficiently. We further show its strengths in uncertainty quantification and guiding exploration in model-based RL. Further information and code is available on the project website[1].

## 1 INTRODUCTION

Deep learning (Goodfellow et al., 2016) has become the cornerstone of contemporary artificial intelligence, proving remarkably effective with large-scale and complex data, such as images. On the other hand, Gaussian processes (GPs, Rasmussen & Williams, 2006), although limited to simpler data, offer functionalities beneficial in sequential learning paradigms, such as continual learning (CL, Parisi et al., 2019; De Lange et al., 2021), reinforcement learning (RL, Sutton & Barto, 2018; Deisenroth & Rasmussen, 2011), and Bayesian optimization (BO, Garnett, 2023). In CL, the challenge is preventing forgetting over the lifelong learning horizon when access to previous data is lost (McCloskey & Cohen, 1989). Notably, a GP's function space provides a more effective representation for CL than an NN's weight space. In RL and BO, GPs provide principled uncertainty estimates that balance the exploration–exploitation trade-off. Furthermore, GPs do not require retraining weights from scratch when incorporating new data.

While GPs could offer several advantages over NNs for sequential learning, they struggle to scale to large and complex data sets, which are common in the real world. In essence, NNs and GPs have complementary strengths that we aim to combine. Previous approaches for converting trained NNs to GPs (Khan et al., 2019; Immer et al., 2021b) show limited applicability to real-world sequential learning problems as they *(i)* rely on subset approximations which do not scale to large data sets and *(ii)* can only incorporate new data by retraining the NN from scratch.

In this paper, we establish a connection between trained NNs and a dual parameterization of GPs (Csató & Opper, 2002; Adam et al., 2021; Chang et al., 2023), which is favourable for sequential learning. In contrast to previous work that utilizes subsets of training data (Immer et al., 2021a), our dual parameterization allows us to sparsify the representation whilst capturing the contributions from *all* data points, essential for predictive uncertainty. We refer to our method as Sparse Function-space Representation (SFR)—a sparse GP derived from a trained NN. A preliminary version of SFR was presented in Scannell et al. (2023). We show how SFR's dual parameterization *(i)* maintains a representation of previous data in CL, *(ii)* incorporates new data without needing to retrain the NN (see Fig. 1), and *(iii)* balances the exploration–exploitation trade-off in sequential decision-making

---

[*]Equal contribution.
[1]`https://aaltoml.github.io/sfr`

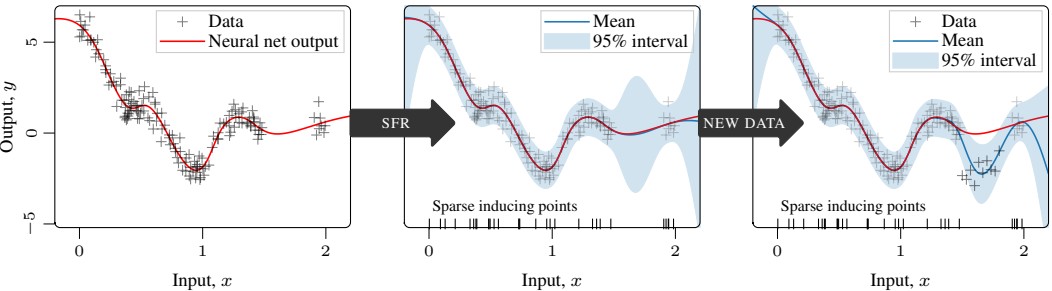

Figure 1: **Regression example with an MLP:** Left: Predictions from the trained neural network. Middle: Our approach summarizes all the training data at the inducing points. The model captures the predictive mean and uncertainty, and (right) incorporates new data without retraining the model.

(RL), via predictive uncertainty. Importantly, our results show that SFR scales to data sets with rich inputs (*e.g.*, images) and millions of data points.

**Related work** Probabilistic methods in deep learning (Neal, 1995; Wilson, 2019) have recently gained increasing attention in the machine learning community as a means for uncertainty quantification. Calculating the posterior distribution of a Bayesian neural network (BNN) is usually intractable. It calls for approximate inference techniques, such as variational inference (Blei et al., 2017), deep ensembles (Lakshminarayanan et al., 2017) and MC dropout (Gal & Ghahramani, 2016)—each with its own strengths and weaknesses. A common approach for uncertainty quantification in NNs is the Laplace-GGN approximation (Daxberger et al., 2021), which takes a trained NN and linearizes it around the optimal weights. The Hessian is approximated using the generalized Gauss–Newton approximation (GGN, Botev et al., 2017). The resulting linear model, with respect to the weights, can be used to obtain uncertainty estimates and refine the NN predictions (Immer et al., 2021a). Immer et al. (2021a) extend Khan et al. (2019) to show the GP connection of the Laplace approximation. However, it does not scale well as it cannot incorporate all data points for uncertainty estimates. Recent work from Ortega et al. (2023) attempts to form a more principled variational sparse GP but resorts to running a second optimization loop. In this paper, we show that such an optimization is not necessary when taking a dual parameterization as shown in Adam et al. (2021) and how such a parameterization lends itself to sequential learning (Chang et al., 2023).

Recent methods in CL, such as FRCL (Titsias et al., 2020), FROMP (Pan et al., 2020), DER (Buzzega et al., 2020), and S-FSVI (Rudner et al., 2022) (see Parisi et al., 2019; De Lange et al., 2021, for an in depth overview) rely on functional regularization methods to alleviate catastrophic forgetting. These approaches obtain state-of-the-art performances among the objective-based techniques in several CL benchmarks compared to their weight-space counterparts: Online-EWC (Schwarz et al., 2018), and VCL (Nguyen et al., 2018). In contrast to the method presented in this paper, these functional regularization methods lack a principled way to incorporate information from all data. Instead, they rely on subset approximations (see App. B.1 for more related work on CL).

## 2 BACKGROUND

We consider supervised learning with inputs $\boldsymbol{x}_i \in \mathbb{R}^D$ and outputs $\boldsymbol{y}_i \in \mathbb{R}^C$ (*e.g.*, regression) or $\boldsymbol{y}_i \in \{0,1\}^C$ (*e.g.*, classification), giving a data set $\mathcal{D} = \{(\boldsymbol{x}_i, \boldsymbol{y}_i)\}_{i=1}^N$. We introduce a NN $f_{\boldsymbol{w}} : \mathbb{R}^D \to \mathbb{R}^C$ with weights $\boldsymbol{w} \in \mathbb{R}^P$ and use a likelihood function $p(\boldsymbol{y}_i \,|\, f_{\boldsymbol{w}}(\boldsymbol{x}_i))$ to link the function values to the output $\boldsymbol{y}_i$ (*e.g.*, categorical for classification). For notational conciseness, we stick to scalar outputs $y_i$ and denote sets of inputs and outputs as $\boldsymbol{X}$ and $\boldsymbol{y}$, respectively.

**BNNs** In Bayesian deep learning, we place a prior over the weights $p(\boldsymbol{w})$ and aim to calculate the posterior over the weights given the data $p(\boldsymbol{w} \,|\, \mathcal{D})$. Given the weight posterior $p(\boldsymbol{w} \,|\, \mathcal{D})$, BNNs make probabilistic predictions $p_{\text{BNN}}(y_i \,|\, \boldsymbol{x}_i, \mathcal{D}) = \mathbb{E}_{p(\boldsymbol{w} \,|\, \mathcal{D})} [p(y_i \,|\, f_{\boldsymbol{w}}(\boldsymbol{x}_i))]$. The posterior $p(\boldsymbol{w} \,|\, \mathcal{D}) \propto p(\boldsymbol{y} \,|\, f_{\boldsymbol{w}}(\boldsymbol{X})) \, p(\boldsymbol{w})$ is generally not available in closed form so we resort to approximations.

**MAP** It is common to train NN weights $\boldsymbol{w}$ to minimize the (regularized) empirical risk,

$$\boldsymbol{w}^* = \arg\min_{\boldsymbol{w}} \underbrace{\mathcal{L}(\mathcal{D}, \boldsymbol{w})}_{-\log p(\mathcal{D}, \boldsymbol{w})} = \arg\min_{\boldsymbol{w}} \sum_{i=1}^N \underbrace{\ell(f_{\boldsymbol{w}}(\boldsymbol{x}_i), y_i)}_{-\log p(y_i \,|\, f_{\boldsymbol{w}}(\boldsymbol{x}_i))} + \underbrace{\mathcal{R}(\boldsymbol{w})}_{-\log p(\boldsymbol{w})}, \quad (1)$$

where the objective $\mathcal{L}(\mathcal{D}, \boldsymbol{w})$ corresponds to the negative log-joint distribution $-\log p(\mathcal{D}, \boldsymbol{w})$ as the loss $\ell(f_{\boldsymbol{w}}(\boldsymbol{x}_i), y_i)$ can be interpreted as a negative log-likelihood $-\log p(y_i \mid f_{\boldsymbol{w}}(\boldsymbol{x}_i))$ and the regularizer $\mathcal{R}(\boldsymbol{w})$ corresponds to a negative log-prior $-\log p(\boldsymbol{w})$. For example, a weight decay regularizer $\mathcal{R}(\boldsymbol{w}) = \frac{\delta}{2}\|\boldsymbol{w}\|_2^2$ corresponds to a Gaussian prior $p(\boldsymbol{w}) = \mathcal{N}(\boldsymbol{w} \mid \boldsymbol{0}, \delta^{-1}\boldsymbol{I})$, with prior precision $\delta$. As such, we can view Eq. (1) as the maximum *a posteriori* (MAP) solution.

**Laplace approximation** The Laplace approximation (MacKay, 1992; Daxberger et al., 2021) builds upon this objective and approximates the weight posterior around the MAP weights $(\boldsymbol{w}^*)$ by setting the covariance to the Hessian of the posterior,

$$p(\boldsymbol{w} \mid \mathcal{D}) \approx q(\boldsymbol{w}) = \mathcal{N}(\boldsymbol{w} \mid \boldsymbol{w}^*, \boldsymbol{\Sigma}) \quad \text{with} \quad \boldsymbol{\Sigma} = -\left[\nabla^2_{\boldsymbol{w}\boldsymbol{w}} \log p(\boldsymbol{w} \mid \mathcal{D})|_{\boldsymbol{w}=\boldsymbol{w}^*}\right]^{-1}. \quad (2)$$

Computing this requires calculating the Hessian of the log-likelihood from Eq. (1). The GGN approximation is used extensively to ensure positive semi-definiteness,

$$\nabla^2_{\boldsymbol{w}\boldsymbol{w}} \log p(\boldsymbol{y} \mid f_{\boldsymbol{w}}(\boldsymbol{X})) \stackrel{\text{GGN}}{\approx} \boldsymbol{J}_{\boldsymbol{w}}(\boldsymbol{X})^\top \nabla^2_{\boldsymbol{f}\boldsymbol{f}} \log(\boldsymbol{y} \mid \boldsymbol{f}) \boldsymbol{J}_{\boldsymbol{w}}(\boldsymbol{X}) \quad \text{and} \quad \boldsymbol{J}_{\boldsymbol{w}}(\boldsymbol{x}) := [\nabla_{\boldsymbol{w}} f_{\boldsymbol{w}}(\boldsymbol{x})]^\top, \quad (3)$$

where $\boldsymbol{f} = f_{\boldsymbol{w}}(\boldsymbol{X})$ denotes set of function values at the training inputs. In practice, the diagonal (Khan et al., 2018; Graves, 2011) or Kronecker factorization (Ritter et al., 2018) of the GGN are used for scalablity. Immer et al. (2021b) highlighted that the GGN approximation corresponds to a local linearization of the NN, showing predictions can be made with a generalized linear model (GLM),

$$p_{\text{GLM}}(y_i \mid \boldsymbol{x}_i, \mathcal{D}) = \mathbb{E}_{q(\boldsymbol{w})}\left[p(y_i \mid f_{\boldsymbol{w}^*}^{\text{lin}}(\boldsymbol{x}_i))\right] \quad \text{with} \quad f_{\boldsymbol{w}^*}^{\text{lin}}(\boldsymbol{x}) = f_{\boldsymbol{w}}(\boldsymbol{x}) + \boldsymbol{J}_{\boldsymbol{w}^*}(\boldsymbol{x})(\boldsymbol{w} - \boldsymbol{w}^*). \quad (4)$$

**GPs** As Gaussian distributions remain tractable under linear transformations, we can convert the linear model from weight space to function space (see Ch. 2.1 in Rasmussen & Williams, 2006). As shown in Immer et al. (2021b), the Bayesian GLM in Eq. (4) has an equivalent GP formulation,

$$p_{\text{GP}}(y_i \mid \boldsymbol{x}_i, \mathcal{D}) = \mathbb{E}_{q(f_i)}\left[p(y_i \mid f_i)\right] \quad \text{and} \quad q(f_i) = \mathcal{N}\left(f_{\boldsymbol{w}^*}(\boldsymbol{x}_i), k_{ii} - \boldsymbol{k}_{\boldsymbol{x}i}^\top(\boldsymbol{K}_{\boldsymbol{x}\boldsymbol{x}} + \boldsymbol{\Lambda}^{-1})^{-1}\boldsymbol{k}_{\boldsymbol{x}i}\right), \quad (5)$$

where the kernel $\kappa(\boldsymbol{x}, \boldsymbol{x}') = \frac{1}{\delta}\boldsymbol{J}_{\boldsymbol{w}^*}(\boldsymbol{x})\,\boldsymbol{J}_{\boldsymbol{w}^*}^\top(\boldsymbol{x}')$ is the Neural Tangent Kernel (NTK, Jacot et al., 2018), $f_i = f_{\boldsymbol{w}}(\boldsymbol{x}_i)$ is the function output at $\boldsymbol{x}_i$, $\boldsymbol{\Lambda} = -\nabla^2_{\boldsymbol{f}\boldsymbol{f}} \log p(\boldsymbol{y} \mid \boldsymbol{f})$ can be interpreted as per-input noise, the $ij^{\text{th}}$ entry of matrix $\boldsymbol{K}_{\boldsymbol{x}\boldsymbol{x}} \in \mathbb{R}^{N \times N}$ is $\kappa(\boldsymbol{x}_i, \boldsymbol{x}_j)$, $\boldsymbol{k}_{\boldsymbol{x}i}$ is a vector where each $j^{\text{th}}$ element is $\kappa(\boldsymbol{x}_i, \boldsymbol{x}_j)$, and $k_{ii} = \kappa(\boldsymbol{x}_i, \boldsymbol{x}_i)$.

**Sparse GPs** The GP formulation in Eq. (5) requires inverting an $N \times N$ matrix which has complexity $\mathcal{O}(N^3)$. This limits its applicability to large data sets, which are common in deep learning. Sparse GPs reduce the computational complexity by representing the GP as a low-rank approximation at a set of inducing inputs $\boldsymbol{Z} = [\boldsymbol{z}_1, \ldots, \boldsymbol{z}_M]^\top \in \mathbb{R}^{M \times D}$ with corresponding inducing variables $\boldsymbol{u} = f(\boldsymbol{Z})$ (see Quiñonero-Candela & Rasmussen, 2005, for an early overview). The approach by Titsias (2009) (also used in the DTC approximation), defines the marginal predictive distribution as $q_{\boldsymbol{u}}(f_i) = \int p(f_i \mid \boldsymbol{u}) q(\boldsymbol{u}) \, \mathrm{d}\boldsymbol{u}$ with $q(\boldsymbol{u}) = \mathcal{N}(\boldsymbol{u} \mid \boldsymbol{m}, \boldsymbol{S})$ (as in Titsias, 2009; Hensman et al., 2013). The sparse GP predictive posterior is

$$\mathbb{E}_{q_{\boldsymbol{u}}(f_i)}[f_i] = \boldsymbol{k}_{\boldsymbol{z}i}^\top \boldsymbol{K}_{\boldsymbol{z}\boldsymbol{z}}^{-1}\boldsymbol{m} \quad \text{and} \quad \text{Var}_{q_{\boldsymbol{u}}(f_i)}[f_i] = k_{ii} - \boldsymbol{k}_{\boldsymbol{z}i}^\top(\boldsymbol{K}_{\boldsymbol{z}\boldsymbol{z}}^{-1} - \boldsymbol{K}_{\boldsymbol{z}\boldsymbol{z}}^{-1}\boldsymbol{S}\boldsymbol{K}_{\boldsymbol{z}\boldsymbol{z}}^{-1})\boldsymbol{k}_{\boldsymbol{z}i}, \quad (6)$$

where $\boldsymbol{K}_{\boldsymbol{z}\boldsymbol{z}}$ and $\boldsymbol{k}_{\boldsymbol{z}i}$ are defined similarly to $\boldsymbol{K}_{\boldsymbol{x}\boldsymbol{x}}$ and $\boldsymbol{k}_{\boldsymbol{x}i}$ but over the inducing points $\boldsymbol{Z}$. We have assumed a zero mean function. Note that the parameters ($\boldsymbol{m}$ and $\boldsymbol{S}$) are usually obtained via variational inference, which requires further optimization.

The GP formulation from Immer et al. (2021b), shown in Eq. (5), struggles to scale to large data sets and it cannot incorporate new data by conditioning on it because its posterior mean is the NN $f_{\boldsymbol{w}^*}(\boldsymbol{x})$. Our method overcomes both of these limitations via a dual parameterization, which enables us to *(i)* sparsify the GP without further optimization, and *(ii)* incorporate new data without retraining.

## 3 SFR: SPARSE FUNCTION-SPACE REPRESENTATION OF NNS

In this section, we present our method, named SFR, which converts a trained NN into a GP (see Fig. 2 for an overview). SFR is built upon a dual parameterization of the GP posterior. That is, in contrast to previous approaches, which adopt the GP formulation in Eq. (5), we use a dual parameterization consisting of parameters $\boldsymbol{\alpha}$ and $\boldsymbol{\beta}$, which gives rise to the predictive posterior,

$$\mathbb{E}_{p(f_i \mid \boldsymbol{y})}[f_i] = \boldsymbol{k}_{\boldsymbol{x}i}^\top \boldsymbol{\alpha} \quad \text{and} \quad \text{Var}_{p(f_i \mid \boldsymbol{y})}[f_i] = k_{ii} - \boldsymbol{k}_{\boldsymbol{x}i}^\top(\boldsymbol{K}_{\boldsymbol{x}\boldsymbol{x}} + \text{diag}(\boldsymbol{\beta})^{-1})^{-1}\boldsymbol{k}_{\boldsymbol{x}i}. \quad (7)$$

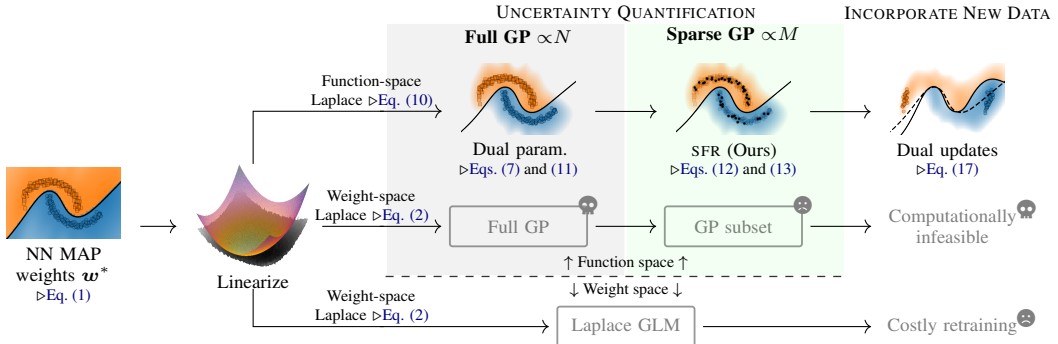

Figure 2: **SFR overview:** We linearize the trained NN around the MAP weights $\boldsymbol{w}^*$ and interpret in function space, via a kernel formulation $\kappa(\cdot, \cdot)$ (Eq. (9)). In contrast to previous approaches, we perform a Laplace approximation on the function-space objective Eq. (10). This leads to SFR's dual parameterization, scaling to large data sets Eq. (12) and incorporating new data efficiently Eq. (17).

See App. A.1 for background. Eq. (7) states that the first two moments of the resultant posterior process (not restricted to GPs), can be parameterized via the dual parameters $\boldsymbol{\alpha}, \boldsymbol{\beta} \in \mathbb{R}^N$, defined as

$$\alpha_i := \mathbb{E}_{p(f_i \mid \boldsymbol{y})}[\nabla_f \log p(y_i \mid f)|_{f=f_i}] \quad \text{and} \quad \beta_i := -\mathbb{E}_{p(f_i \mid \boldsymbol{y})}[\nabla^2_{ff} \log p(y_i \mid f_i)|_{f=f_i}]. \quad (8)$$

Eq. (8) holds for generic likelihoods and involves no approximations since the expectation is under the exact posterior, given that the model can be expressed in a kernel formulation. Eq. (7) and Eq. (8) highlight that the approximate inference technique, usually viewed as a posterior approximation, can alternatively be interpreted as an approximation of the expectation of loss (likelihood) gradients.

**Linear model for dual updates** Previous NN to GP methods (Khan et al., 2019; Immer et al., 2021b) set the GP mean to be the NN's prediction (Eq. (5)). In contrast, we linearize as

$$f_{\boldsymbol{w}^*}(\boldsymbol{x}) \approx \boldsymbol{J}_{\boldsymbol{w}^*}(\boldsymbol{x})\, \boldsymbol{w}^* \quad \implies \mu(\boldsymbol{x}) = 0 \quad \text{and} \quad \kappa(\boldsymbol{x}, \boldsymbol{x}') = \frac{1}{\delta} \boldsymbol{J}_{\boldsymbol{w}^*}(\boldsymbol{x})\, \boldsymbol{J}^\top_{\boldsymbol{w}^*}(\boldsymbol{x}'), \quad (9)$$

because this leads to a GP with a zero mean function. Importantly, this formulation enables us to incorporate new data via our dual parameterization, as we detail in Sec. 4.

**Dual parameters from NN** Eq. (9) gives us a way to convert a weight-space NN into function space. However, we go a step further and convert the weight-space objective in Eq. (1) to function space, *i.e.* $\mathcal{L}(\mathcal{D}, \boldsymbol{w}) = \sum_{i=1}^N \log p(y_i \mid f_i) + \log p(\boldsymbol{f})$. We can then approximate the function-space posterior $q(\boldsymbol{f}) = \mathcal{N}(\boldsymbol{f} \mid \boldsymbol{m_f}, \boldsymbol{S_f})$ by applying the Laplace approximation to this function-space objective. That is, we can obtain $\boldsymbol{m_f}$ and $\boldsymbol{S_f}$ from the stationary point of the objective:

$$\boldsymbol{0} = \nabla_f \log p(y_i \mid f)|_{f=f_i} - \boldsymbol{K}^{-1}_{\boldsymbol{xx}} \boldsymbol{m_f} \quad \text{and} \quad \boldsymbol{S}^{-1}_{\boldsymbol{f}} = -\nabla^2_{ff} \log p(y_i \mid f)|_{f=f_i} + \boldsymbol{K}^{-1}_{\boldsymbol{xx}}. \quad (10)$$

Comparing Eq. (10) and Eq. (7), we can see that our approximate inference technique (Laplace) has simplified the dual parameter calculation in Eq. (8), *i.e.* the expectation is now removed, meaning:

$$\hat{\alpha}_i := \nabla_f \log p(y_i \mid f)|_{f=f_i} \quad \text{and} \quad \hat{\beta}_i := -\nabla^2_{ff} \log p(y_i \mid f)|_{f=f_i}. \quad (11)$$

The variables in Eq. (11) are easily computable using the NN MAP, *i.e.* $f_i = f_{\boldsymbol{w}^*}(\boldsymbol{x}_i)$. Note that if we had a weight-space posterior $q(\boldsymbol{w})$ we could use the law of unconscious statistician (pushforward measure) and replace the expectations in Eq. (8) with $q(\boldsymbol{w})$. As such, our method can be applied to any weight-space approximate inference technique. See App. A.2 for details on the dual parameters for different losses. Substituting Eq. (11) into Eq. (7), we obtain our GP based on the trained NN. Predicting with Eq. (7) costs $\mathcal{O}(N^3)$, which limits its applicability on large data sets.

**Sparsification via dual parameters** Sparse GPs reduce the computational complexity by representing the GP as a low-rank approximation induced by a sparse set of inducing points (see Quiñonero-Candela & Rasmussen, 2005, for an early overview). In the general setting, with non-Gaussian likelihoods, it is not obvious how to sparsify the full GP in Eq. (5). Previous approaches have resorted to *(i)* simply selecting a subset (Immer et al., 2021b) and *(ii)* performing variational inference, *i.e.* running a second optimization loop (Ortega et al., 2023). The dual parameter formulation in Eq. (7) enables us leverage any sparsification method. Notably, we do not need to resort to selecting

a subset or performing a second optimization loop. In this work, we opt for the approach suggested by Titsias (2009); Hensman et al. (2013), shown in Eq. (6). However, instead of parameterizing a variational distribution as $q(\boldsymbol{u}) = \mathcal{N}(\boldsymbol{u} \,|\, \boldsymbol{m}, \boldsymbol{S})$, we follow the insights from Adam et al. (2021), that the posterior under this model bears a structure akin to Eq. (7). As such, we project the dual parameters onto the inducing points giving us sparse dual parameters. Using this sparse definition of the dual parameters, our sparse GP posterior is given by

$$\mathbb{E}_{q_{\boldsymbol{u}}(\boldsymbol{f})}[f_i] = \boldsymbol{k}_{\boldsymbol{z}i}^\top \boldsymbol{K}_{\boldsymbol{zz}}^{-1} \boldsymbol{\alpha}_{\boldsymbol{u}} \quad \text{and} \quad \mathrm{Var}_{q_{\boldsymbol{u}}(\boldsymbol{f})}[f_i] = k_{ii} - \boldsymbol{k}_{\boldsymbol{z}i}^\top [\boldsymbol{K}_{\boldsymbol{zz}}^{-1} - (\boldsymbol{K}_{\boldsymbol{zz}} + \boldsymbol{B}_{\boldsymbol{u}})^{-1}]\boldsymbol{k}_{\boldsymbol{z}i}, \quad (12)$$

with sparse dual parameters,

$$\boldsymbol{\alpha}_{\boldsymbol{u}} = \sum_{i=1}^N \boldsymbol{k}_{\boldsymbol{z}i}\, \hat{\alpha}_i \in \mathbb{R}^M \quad \text{and} \quad \boldsymbol{B}_{\boldsymbol{u}} = \sum_{i=1}^N \boldsymbol{k}_{\boldsymbol{z}i}\, \hat{\beta}_i\, \boldsymbol{k}_{\boldsymbol{z}i}^\top \in \mathbb{R}^{M \times M}. \quad (13)$$

In our experiments, we sampled the inducing inputs $\boldsymbol{Z} = [\boldsymbol{z}_1, \ldots, \boldsymbol{z}_M]^\top \in \mathbb{R}^{M \times D}$ from the training inputs $\boldsymbol{X}$. Note that the sparse dual parameters are now a sum over *all data points*. Contrasting Eq. (12) and Eq. (7), we can see that the computational complexity went down from $\mathcal{O}(N^3)$ to $\mathcal{O}(M^3)$, with $M \ll N$. Crucially, our sparse dual parameterization (Eq. (13)) is a compact representation of the full model projected using the kernel. Alg. A1 details how we compute SFR's dual parameters and App. A.3 analyzes SFR's computational complexity.

We highlight that SFR differs from previous NN to GP methods (Khan et al., 2019; Immer et al., 2021b). To the best of our knowledge, we are the first to formulate a dual GP from a NN. SFR's dual parameterization has two main benefits: *(i)* it enables us to construct a sparse GP (capturing information from all data points) which does not require further optimization, and *(ii)* it enables us to incorporate new data without retraining by conditioning on new data (using Eq. (13), see Eq. (17)).

## 4 SFR FOR SEQUENTIAL LEARNING

In this section, we show how SFR's sparse dual parameterization can equip NNs with important functionalities for sequential learning: *(i)* maintaining a representation of the NN for CL and *(ii)* incorporating new data without retraining from scratch.

**Continual learning** In the continual learning (CL) setting, training is divided into $T$ tasks, each with its own training data set $\mathcal{D}_t = \{(\boldsymbol{x}_i, \boldsymbol{y}_i)\}_{i=1}^{N_t}$. Once a task $t \in \{1, \ldots, T\}$ is complete its data $\mathcal{D}_t$ cannot be accessed in the subsequent tasks, *i.e.*, it is discarded. CL methods based on rehearsal and function-space regularization typically keep a subset of each task's training data to help alleviate forgetting (Buzzega et al., 2020; Pan et al., 2020; Rudner et al., 2022). They show better performance than their weight space equivalents and report state-of-the-art results on CL benchmarks. Previous approaches attempt to perform function-space VCL (Rudner et al., 2022; Pan et al., 2020), however, in order to scale to large data sets, they are forced to approximate the posterior at a subset of the training data, ignoring information from the rest of the training data. In contrast, a sparse function-space VCL method from the GP literature (Chang et al., 2023) uses the sparse approximation to scale using the ELBO at the current iteration,

$$q^*(\boldsymbol{u}) = \arg\max_{q(\boldsymbol{u})} \mathcal{L}_{\mathrm{VCL}} = \arg\max_{q(\boldsymbol{u})} \sum_{i=1}^N \mathbb{E}_{q_{\boldsymbol{u}}(f_i)}[\log p(y_i \,|\, f_i)] - \mathrm{KL}[q(\boldsymbol{u}) \,\|\, q_{\mathrm{old}}(\boldsymbol{u})], \quad (14)$$

where $q_{\mathrm{old}}(\boldsymbol{u})$ is the posterior from the previous iteration. Using SFR we can approximate sparse function-space VCL (see App. B.2 for details). For task $t$, the regularized objective is

$$\boldsymbol{w}_t^* = \arg\min_{\boldsymbol{w}} \mathcal{L}(\mathcal{D}_t, \boldsymbol{w}) + \tau \underbrace{\frac{1}{2} \sum_{s=1}^{t-1} \frac{1}{M} \left\| f_{\boldsymbol{w}_s^*}(\boldsymbol{Z}_s) - f_{\boldsymbol{w}}(\boldsymbol{Z}_s) \right\|_{\bar{\boldsymbol{B}}_s^{-1}}}_{\mathcal{R}_{\mathrm{SFR}}(\boldsymbol{w}, \mathcal{M}_{t-1})}, \quad (15)$$

where $\tau \in \mathbb{R}$ is a hyperparameter that scales the influence of the regularizer, $\boldsymbol{w}_t^*$ denotes the MAP weights from task $t$, and $\boldsymbol{Z}_t \in \mathbb{R}^{M \times D}$ denotes a set of task-specific inducing inputs selected from $\mathcal{D}_t$. For notational convenience, we restrict ourselves to single-output NNs and refer the reader to App. B.3 for details on the multi-output setting. The regularizer resembles a Mahalanobis distance with covariance matrix $\bar{\boldsymbol{B}}_t^{-1}$ given by

$$\bar{\boldsymbol{B}}_t^{-1} = \boldsymbol{K}_{\boldsymbol{zz}}^{-1} \boldsymbol{B}_{\boldsymbol{u}} \boldsymbol{K}_{\boldsymbol{zz}}^{-1} \in \mathbb{R}^{M \times M}, \quad \text{s.t.} \quad \boldsymbol{B}_{\boldsymbol{u}} = \sum_{i \in \mathcal{D}_t} \boldsymbol{k}_{\boldsymbol{z}i}\, \hat{\beta}_i\, \boldsymbol{k}_{\boldsymbol{z}i}^\top, \quad (16)$$

where $\boldsymbol{K_{zz}}$ and $\boldsymbol{k_{zi}}$ are the Gram matrix and the vector computed on the task-specific inducing points $\boldsymbol{Z}_t$. In practice, we randomly select a set of $M$ task-specific inducing inputs $\boldsymbol{Z}_t$ from the previous task's data set $\mathcal{D}_t$. Given these inducing inputs, we then calculate the corresponding inducing variables $\boldsymbol{u}_t = f_{\boldsymbol{w}_t^*}(\boldsymbol{Z}_t)$. Finally, we calculate the regularization matrix Eq. (16) and add all of these entries to a memory buffer, $\mathcal{M}_{t-1} = \{(\boldsymbol{Z}_s, \boldsymbol{u}_s, \bar{\boldsymbol{B}}_s^{-1})\}_{s=1}^{t-1}$, to summarize previous tasks.

Intuitively, the regularizer in Eq. (15) keeps the function values of the NN $f_{\boldsymbol{w}}(\boldsymbol{Z}_s)$ close to those at the MAP of previous tasks $\{\boldsymbol{u}_s\}_{s=1}^{t-1} = \{f_{\boldsymbol{w}_s^*}(\boldsymbol{Z}_s)\}_{s=1}^{t-1}$. It is worth noting that the covariance structure in $\bar{\boldsymbol{B}}_t^{-1}$ relaxes the regularization where there is no training data. Eq. (16) sidesteps computing the full posterior covariance for each task's dataset $\mathcal{D}_t$ as it leverages SFR's dual parameterization to efficiently encode the information from $\mathcal{D}_t$. As such, it offers a scalable solution whilst capturing information from the entire data set $\mathcal{D}_t$.

**Incorporating new data without retraining**  Incorporating new data $\mathcal{D}_{\text{new}}$ into a trained NN is not trivial. Existing approaches typically consider a weight space formulation and directly update the NN's weights (Kirsch et al., 2022; Spiegelhalter & Lauritzen, 1990). In contrast, GPs defined in the function space can incorporate new data easily (Chang et al., 2023). Given our sparse dual parameters (Eq. (13)), we can incorporate new data into SFR with *dual updates*,

$$\boldsymbol{\alpha_u} \leftarrow \boldsymbol{\alpha_u} + \sum_{i \in \mathcal{D}_{\text{new}}} \boldsymbol{k_{zi}} \hat{\alpha}_i \quad \text{and} \quad \boldsymbol{B_u} \leftarrow \boldsymbol{B_u} + \sum_{i \in \mathcal{D}_{\text{new}}} \boldsymbol{k_{zi}} \hat{\beta}_i \boldsymbol{k_{zi}}^\top. \tag{17}$$

See Fig. 1 for an example of SFR incorporating new data and see Alg. A2 and App. A.3 for details of the *dual updates* and their computational complexity. Importantly, SFR's dual parameterization enabled us to incorporate new data *(i)* efficiently and *(ii)* when access to previous data is lost.

## 5 EXPERIMENTS

We present a series of experiments specifically designed to showcase the power of SFR's dual parameterization. We first provide supervised learning experiments (Sec. 5.1) to highlight that SFR's sparse dual parameterization scales to data sets with *(i)* image inputs and *(ii)* over 1 million data points. As a sanity check, we also show that SFR's uncertainty quantification generally matches (or is better than) other *post hoc* BNN methods, such as the Laplace approximation. We then provide sequential learning experiments (Sec. 5.2) which show that *(i)* SFR's sparse dual parameterization can help retain knowledge from previous tasks in CL and *(ii)* SFR's dual updates can incorporate new data fast. We implement all methods in PyTorch (Paszke et al., 2019) and use a GPU cluster. We provide full experiment details in App. D. For illustrative purposes, we show SFR on 1D regression (Fig. 1) and on the 2D BANANA classification task (Fig. A5). Fig. 1 middle shows a trained MLP being converted into SFR and right shows new data being incorporated with the *dual updates* (see Sec. 4). In App. E we visualize SFR's posterior covariance structure on the 1D regression problem.

### 5.1 SUPERVISED LEARNING EXPERIMENTS

This section aims to demonstrate the efficacy of SFR at quantifying uncertainty in several regression and classification tasks and test its performance in dealing with complex and large-scale data sets.

**Experiment setup**  We evaluate the effectiveness of SFR's sparse dual parameterization on eight UCI (Dua & Graff, 2017) classification tasks, two image classification tasks: Fashion-MNIST (FMNIST, Xiao et al., 2017) and CIFAR-10 (Krizhevsky et al., 2009), and the large-scale House-Electric data set. We used a two-layer MLP with width 50 and `tanh` activation functions for the UCI experiments. We used a CNN architecture for the image classification tasks and a wider and deeper MLP for HouseElectric. See App. D.1.2 for full experiment details.

**Baselines**  As SFR represents information from all the training data at a set of inducing points, we compare it to making GP predictions with a subset of the training data (GP subset) on UCI and the image data sets. Note that for a given number of inducing points, the complexity of making predictions with the GP subset matches SFR. As such, the GP subset acts as a baseline whose predictive performance we want to match whilst using fewer inducing points. As a sanity check, we also compare SFR to the Laplace approximation (Laplace PyTorch, Daxberger et al., 2021) when making predictions with *(i)* the nonlinear NN (BNN), *(ii)* the generalised linear model (GLM) in Eq. (4), and *(iii)* the GP predictive from Immer et al. (2021b), the generalization of DNN2GP (Khan

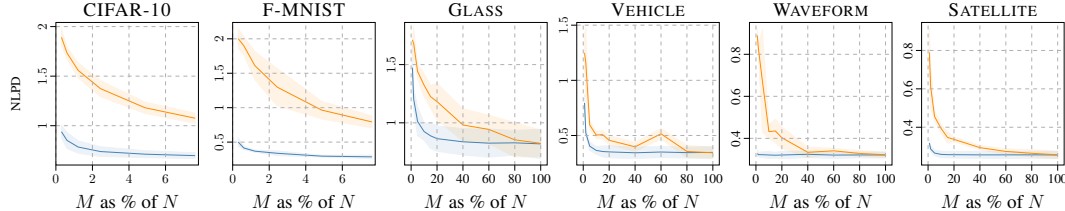

Figure 3: **Effective sparsification:** Comparison of convergence in number of inducing points $M$ in NLPD (mean±std over 5 seeds) on classification tasks: SFR (——) vs. GP subset (——). Our SFR converges fast for all cases showing clear benefits of its ability to summarize all the data in a sparse model.

et al., 2019) to non-Gaussian likelihoods. As GP predictive makes predictions around the NN, we also include results for SFR making predictions with the NN (SFR-NN), instead of the GP mean. Note that the LAPLACE DIAG and LAPLACE KRON experiments use the diagonal (Graves, 2011) or Kronecker factorization (Martens & Grosse, 2015) of the GGN respectively.

**SFR's sparsification** Fig. 3 compares how the predictive performance in terms of negative log predictive density (NLPD, lower is better) of SFR and the GP subset deteriorates as the number of inducing points is lowered from $M = 100\%$ of $N$ to $M = 1\%$ of $N$. SFR is able to summarize the full data set more effectively than the GP subset method as it maintains a good (low) NLPD with fewer inducing points. This demonstrates that the dual parameterization is effective for sparsification.

**SFR on UCI data sets** Tables A4 and A5 further demonstrate SFR's uncertainty quantification on the UCI classification tasks. They show that SFR outperforms the Laplace approximation (when using BNN, GLM and GP predictions) and the GP subset with $M = 20\%$ of $N$ on all eight tasks.

**SFR experiments on image data sets** We evaluate SFR on image data sets, which present a more challenging task and require more complicated NN architectures than UCI experiments. In particular, we report the results obtained using a CNN architecture on FMNIST (Xiao et al., 2017) and CIFAR-10 (Krizhevsky et al., 2009). In Table 1, we report the results obtained keeping the same prior precision $\delta$ at training and inference time. We refer the reader to Apps. D.1.1 and D.1.3 for a detailed explanation of the experiment. The performance of SFR using $M = 2048$ and $M = 3200$ inducing points outperforms the GP subset method and Laplace approximation when using both BNN and GLM predictions. This experiment indicates that SFR's sparse dual parameterization effectively captures information from all data points even when dealing with high-dimensional data.

**Out-of-distribution detection** We follow Ritter et al. (2018); Osawa et al. (2019) and

Table 1: **Image classification results using CNN:** We report NLPD, accuracy and AUROC (mean±std over 5 seeds). SFR outperforms the GP subset and Laplace methods. The prior precision $\delta$ is not tuned *post hoc*.

| | MODEL | $M$ | NLPD ↓ | ACC. (%) ↑ | AUROC ↑ |
|---|---|---|---|---|---|
| F-MNIST | NN MAP | - | 0.23±0.01 | 91.98±0.44 | 0.83±0.05 |
| | LAPLACE DIAG | - | 2.42±0.02 | 10.21±0.66 | 0.50±0.03 |
| | LAPLACE KRON | - | 2.39±0.01 | 9.87±0.66 | 0.51±0.02 |
| | LAPLACE GLM DIAG | - | 1.66±0.02 | 65.19±2.21 | 0.67±0.03 |
| | LAPLACE GLM KRON | - | 1.09±0.04 | 84.79±1.96 | 0.96±0.01 |
| | GP PREDICTIVE | 3200 | 0.47±0.06 | 91.51±0.45 | **0.97**±0.01 |
| | SFR-NN (Ours) | 3200 | **0.31**±0.03 | 91.86±0.40 | **0.97**±0.01 |
| | GP SUBSET | 2048 | 0.97±0.13 | 77.32±8.83 | 0.93±0.03 |
| | | 3200 | 0.79±0.09 | 82.52±4.10 | **0.95**±0.02 |
| | SFR (Ours) | 2048 | **0.30**±0.01 | **91.68**±0.51 | 0.95±0.02 |
| | | 3200 | **0.29**±0.02 | 91.74±0.47 | 0.96±0.01 |
| CIFAR-10 | NN MAP | - | 0.69±0.03 | 77.00±1.04 | 0.85±0.02 |
| | LAPLACE DIAG | - | 2.37±0.05 | 10.08±0.24 | 0.48±0.01 |
| | LAPLACE KRON | - | 2.36±0.01 | 9.78±0.41 | 0.49±0.01 |
| | LAPLACE GLM DIAG | - | 1.33±0.05 | 71.96±1.38 | **0.82**±0.03 |
| | LAPLACE GLM KRON | - | 1.04±0.08 | 75.56±1.63 | 0.64±0.04 |
| | GP PREDICTIVE | 3200 | 0.90±0.02 | 76.07±1.17 | **0.79**±0.02 |
| | SFR-NN (Ours) | 3200 | 0.79±0.02 | 76.59±1.21 | 0.79±0.03 |
| | GP SUBSET | 2048 | 1.18±0.06 | 66.40±3.69 | 0.71±0.05 |
| | | 3200 | 1.08±0.05 | 69.75±3.23 | 0.75±0.01 |
| | SFR (Ours) | 2048 | **0.74**±0.02 | **78.40**±0.83 | **0.79**±0.02 |
| | | 3200 | **0.72**±0.02 | **78.48**±0.98 | 0.79±0.02 |

compare the predictive entropies of the in-distribution (ID) vs. out-of-distribution (OOD) data. We desire low predictive entropy ID to indicate that predictions can be made confidently and high predictive entropy OOD to indicate predictions cannot be made confidently. Fig. 4 shows predictive entropy histograms for models trained on FMNIST and evaluated OOD on MNIST data. SFR has low predictive entropy at the ID data and high predictive entropy at the OOD data, just as we desire. We further quantify SFR's OOD detection using the area under the receiver operating characteristic curve (AUROC), a commonly used threshold-free evaluation metric. Table 1 reports the AUROC metric (higher is better), which further demonstrates that SFR has good OOD performance. See App. D.1.4 for further details and experiments on CIFAR-10.

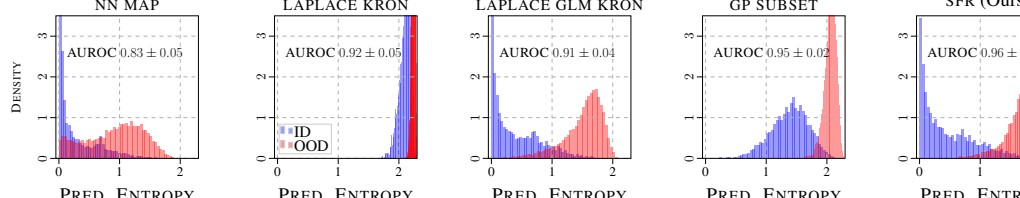

Figure 4: **OOD detection with CNN:** Histograms showing each method's predictive entropy at ID data (FMNIST, blue) where lower is better and at OOD data (MNIST, red) where higher is better.

**SFR scales to large data sets**   It is well known that GPs struggle to scale to large data sets. Nevertheless, Wang et al. (2019) scaled GPs to a million data points. We repeat their main experiment on the HouseElectric data set and recover better NLPD of $-0.153\pm0.001$ vs. $0.024\pm0.984$ for an SVGP model (Hensman et al., 2013) on the same data set with a 5-fold split. Furthermore, we also record a better wall-clock time of $6129\pm996$ s vs. $11982\pm455$ s. This indicates that SFR outperforms GPs on large data sets in terms of both computation time and predictive performance.

## 5.2 SEQUENTIAL LEARNING EXPERIMENTS

The results in Sec. 5.1 motivate using SFR in more challenging sequential learning problems. In this section, we demonstrate the effectiveness of SFR's sparse dual parameterization in sequential learning settings. **Representation:** First, we show that SFR's sparse dual parameterization can help retain knowledge from previous tasks in CL. **Incorporating new data:** We then show that SFR can incorporate new data fast via dual updates. **Uncertainty:** Finally, we demonstrate that SFR's uncertainty estimates are good enough to help guide exploration in model-based RL.

**Continual learning**   The regularizer described in Sec. 4 can be used in CL to retain a compact representation of the NN. We report experiments in the single-head (SH) setting because it is more realistic (and harder) than multi-head (MH) (see discussion in Van de Ven & Tolias (2019)). Our regularizer is evaluated on three CL benchmarks: Split-MNIST (S-MNIST), Split-FashionMNIST (S-FMNIST), and the 10-tasks Permuted-MNIST (P-MNIST). We adhere to the same setups outlined in Rudner et al. (2022); Pan et al. (2020), which use a two-layer MLP with 256 hidden units and ReLU activation for S-MNIST and S-FMNIST and a two-layer MLP with 100 hidden units for P-MNIST, to ensure consistency (see App. D.2 for full details).

Table 2: **Continual learning experiments:** Accuracy$\pm$std, bolding based on a $t$-test. *Methods rely on weight regularization.

| Method | S-MNIST (SH) 40 pts./task | S-MNIST (SH) 200 pts./task | S-FMNIST (SH) 200 pts./task | P-MNIST (SH) 200 pts./task |
|---|---|---|---|---|
| ONLINE-EWC* | $19.95\pm0.28$ | $19.95\pm0.28$ | $19.48\pm0.01$ | $74.87\pm1.81$ |
| SI* | $19.82\pm0.09$ | $19.82\pm0.09$ | $19.80\pm0.21$ | $88.39\pm1.37$ |
| VCL (w\coresets) | $22.31\pm2.00$ | $32.11\pm1.16$ | $53.59\pm3.74$ | $93.08\pm0.11$ |
| DER | $85.26\pm0.54$ | $92.13\pm0.45$ | $\mathbf{82.03}\pm0.57$ | $93.08\pm0.11$ |
| FROMP | $75.21\pm2.05$ | $89.54\pm0.72$ | $78.83\pm0.46$ | $94.90\pm0.04$ |
| S-FSVI | $84.51\pm1.30$ | $92.87\pm0.14$ | $77.54\pm0.40$ | $\mathbf{95.76}\pm0.02$ |
| SFR (Ours) | $\mathbf{89.22}\pm0.76$ | $\mathbf{94.19}\pm0.26$ | $81.96\pm0.24$ | $95.58\pm0.08$ |
| L2 (Ours) abl. | $87.53\pm0.36$ | $93.72\pm0.10$ | $81.21\pm0.36$ | $94.96\pm0.15$ |

We compare our method against two categories of methods: *(i)* weight-regularization methods: Online-EWC (Schwarz et al., 2018), SI (Zenke et al., 2017), and VCL (with coresets) (Nguyen et al., 2018), and *(ii)* function-based regularization methods: DER (Buzzega et al., 2020), FROMP (Pan et al., 2020), and S-FSVI (Rudner et al., 2022) . We also introduce an ablation study where we replace our $\bar{B}_t^{-1}$ with an identity matrix $I_M$, equivalent to having an L2 regularization between current and old function outputs. In Table 2, we use 200 points for each task and we further demonstrate our method's ability to compress the task data set information on a lower number of points for S-MNIST.

From Table 2, it is clear that the weight-space regularization methods fail entirely compared to the function-space methods on S-MNIST and S-FMNIST but are still achieving comparable results on P-MNIST. Considering only function-space methods, we can see that SFR achieves the best results on most data sets and is particularly effective when using fewer points per task. On S-FMNIST, our method obtains close results to DER, which regularizes the model by taking the mean squared error between current and old function outputs without accounting for the covariance structure of the loss, similar to our L2 ablation. However, DER resorts to reservoir sampling (Vitter, 1985) that continuously updates the set of points instead of selecting them at the task boundary. The SFR-based

Table 3: **SFR's dual updates are fast:** Comparison of incorporating new data $\mathcal{D}_2$ via SFR's dual updates (Eq. (17)) vs. retraining from scratch (on $\mathcal{D}_1 \cup \mathcal{D}_2$). SFR's dual updates improve the NLPD (val.+std as bar, lower better) whilst being significantly faster than retraining from scratch.

| | | NLPD ↓ | | | Time (s) ↓ | |
| | Train w. $\mathcal{D}_1$ | Updates w. $\mathcal{D}_2$ | Retrain w. $\mathcal{D}_1 \cup \mathcal{D}_2$ | Train w. $\mathcal{D}_1$ | Updates w. $\mathcal{D}_2$ | Retrain w. $\mathcal{D}_1 \cup \mathcal{D}_2$ |
|---|---|---|---|---|---|---|
| AIRFOIL | 0.60 | 0.50 | 0.47 | 19.65 | 0.04 | 18.22 |
| BOSTON | 0.23 | 0.16 | 0.13 | 11.45 | 0.02 | 7.48 |
| PROTEIN | 0.42 | 0.15 | 0.14 | 30.17 | 0.82 | 30.61 |

regularizer obtains comparable results to the best-performing method on P-MNIST, S-FSVI, which requires heavy variational inference computations compared to SFR.

**Incorporating new data via dual updates**   SFR's dual parameterization enabled us to formulate equations for incorporating new data $\mathcal{D}_2$ into a NN previously trained on $\mathcal{D}_1$ (using Eq. (17)). We test SFR's so-called dual updates on three UCI regression data sets. See App. D.3 for more details on our experimental set-up. Table 3 shows that incorporating new data $\mathcal{D}_2$ via SFR's dual updates is significantly faster than retraining the NN from scratch (on $\mathcal{D}_1 \cup \mathcal{D}_2$) and offers improvements in the NLPD. Whilst SFR's dual updates show benefits in simple regression tasks, our initial experiments on image data sets did not show any improvement in predictive performance. We think this is because the dual updates require the kernel (NTK) to be accurate outside of the training data and this is unlikely because it was not enforced to be stationary. Nevertheless, incorporating new data via dual updates is very fast relative to retraining from scratch. As such, investigating SFR's dual updates in downstream applications, where retraining from scratch is too costly, is worthwhile. For example, batch Bayesian optimization (Wu & Frazier, 2016).

**Reinforcement learning under sparse rewards**   As a final experiment, we demonstrate the capability of SFR to use its uncertainty estimates as guidance for exploration in model-based RL. We use SFR to learn a dynamics model within a model-based RL strategy that employs posterior sampling to guide exploration (Osband & Van Roy, 2017). We use the cartpole swingup task in MuJoCo (Todorov et al., 2012) (see Fig. A7); the goal is to swing the pole up and balance it. We increase the difficulty of exploration by using a sparse reward function. See App. C for an overview of the RL problem, details of the algorithm, and the experiment setup.

Fig. A7 shows training curves for using SFR as the dynamics model (——), along with a Laplace-GGN (Immer et al., 2021a) with GLM predictions (——), an ensemble of NNs (——), and a basic MLP without uncertainty (——). To ensure a fair comparison, we maintain the same MLP architecture/training scheme across all these methods and incorporate them into the same model-based RL algorithm (see App. C). We also compare our results with (DDPG, Lillicrap et al., 2016), a model-free RL algorithm (——). The training curves show that SFR's uncertainty estimates help exploration converge in fewer episodes, demonstrating higher sample efficiency.

## 6   DISCUSSION AND CONCLUSION

We introduced SFR, a novel approach for representing NNs in sparse function space. Our method is built upon a dual parameterization which offers a powerful mechanism for capturing predictive uncertainty, providing a compact representation suitable for CL, and incorporating new data without retraining. SFR is applicable on large data sets with rich inputs (*e.g.*, images), where GPs are known to fail. This is because SFR learns the covariance structure so can learn non-stationary covariance functions, which would otherwise be hard to specify. These aspects were demonstrated in a wide range of problems, data sets, and learning contexts. We showcased SFR's ability to capture uncertainty in UCI and image classification, established its potential for CL, and verified its applicability in RL.

**Limitations**   In practical terms, SFR serves a role similar to a sparse GP. However, unlike vanilla GPs, it does not provide a straightforward method for specifying the prior covariance function. This limitation can be addressed indirectly: the architecture of the NN and the choice of activation functions can be used to implicitly encode the prior assumptions, thereby incorporating a broad range of inductive biases into the model. It is important to note that we linearize the network around the MAP weights $\boldsymbol{w}^*$, resulting in the function-space prior (and consequently the posterior) being only a locally linear approximation of the NN model. As such, when we incorporate new data with Eq. (17) the model becomes outdated. Nevertheless, in some cases we can still improve predictions without retraining.

AUTHOR CONTRIBUTIONS

AJS (Aidan J. Scannell), RM, and PC jointly devised the methodological ideas used in the paper through discussion with JP and AHS. AJS and RM jointly developed the code base. RM conducted the CL experiments, whilst AJS conducted the RL experiments. AJS and RM jointly conducted the supervised learning experiments with help from ET. AJS lead the paper writing with the help of RM and PC. All authors contributed to finalizing the manuscript.

ACKNOWLEDGMENTS

AJS was supported by the Research Council of Finland from the Flagship program: Finnish Center for Artificial Intelligence (FCAI). AHS acknowledges funding from the Research Council of Finland (grant id 339730). We acknowledge CSC – IT Center for Science, Finland, for awarding this project access to the LUMI supercomputer, owned by the EuroHPC Joint Undertaking, hosted by CSC (Finland) and the LUMI consortium through CSC. We acknowledge the computational resources provided by the Aalto Science-IT project. We thank Rui Li, Severi Rissanen, and the rest of the Aalto Machine Learning research group for valuable feedback on our manuscript. Finally, this research was supported by the NVIDIA AI Technology Center (NVAITC) Finland. We thank Niki Loppi of NVAITC Finland for his help with multi-GPU/node implementation and experimenting with float and half precision.

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

## APPENDIX

This appendix is organized as follows. App. A provides a more extensive overview of the technical details of SFR. App. B covers additional details on the SFR-based regularizer used for CL. App. C provides a more extensive write up of the reinforcement learning setup used in the experiments. App. D provides the full details of each individual experiment in the main paper. Finally, App. E provides a visualisation of SFR's covariance matrix.

# A  METHOD DETAILS

In this section, we provide further details on our method. App. A.1 provides further background and derivations of the dual parameters. App. A.2 shows how SFR's dual parameters are computed for the different losses used in our experiments. We then discuss the computational complexity of our method in App. A.3. In App. A.4 we detail considerations for using SFR in practice. In App. A.5 we provide algorithms for *(i)* the computation of SFR's sparse dual parameters and *(ii)* SFR's dual updates.

## A.1  DUAL PARAMETERIZATION: BACKGROUND AND DERIVATION

Any empirical risk minimization objective (that assumes likelihood terms are factorized across data) can be reparameterized in to what Adam et al. (2021) call the dual parameterization. In the support vector machine (SVM) literature, the connection to dual parameterization was pointed out in Chapelle (2007). In the GP literature, they first appeared in Csató & Opper (2002) and then Opper & Archambeau (2009) showed them for the variational GP objective. More recently, Adam et al. (2021) coined the dual parameterization for sparse variational GPs and showed they correspond to the dual Lagrangian optimization objective.

Dual parameters take slightly different forms depending on the types of objective. In particular, whether the objective is Bayesian (*e.g.* ELBO) or non-Bayesian (*e.g.* MAP objective). They are identified from the stationary point of the objective. As we can determine the posterior's original parameters ($m_f$ and $S_f$), given the dual parameters, we follow Adam et al. (2021) and call them dual parameters. We now derive the dual parameters for these two type of objectives.

**Dual parameters for ELBO objective**  Variational inference casts inference to an optimization seeking to maximize the ELBO, given by

$$m_f^*, S_f^* = \arg \max_{m_f, S_f} \sum_{i=1}^{N} \mathbb{E}_{q(f_i)}[\log p(y_i \,|\, f_i)] - \mathrm{KL}[q(f) \,\|\, p(f)], \tag{18}$$

with variational posterior $q(f) = \mathcal{N}(f \,|\, m_f, S_f)$ and prior $p(f) = \mathcal{N}(f \,|\, 0, K)$. Note that $m_f$ and $S_f$ are the variational parameters to be optimized. The stationary point of the ELBO in Eq. (18) is given by

$$K\alpha^* = m_f^*, \qquad (S_f^*)^{-1} = \mathrm{diag}(\beta^*) + K^{-1}, \tag{19}$$

where $q^*(f) = \mathcal{N}(f \,|\, m_f^*, S_f^*)$ and $\alpha^* = [\alpha_0^*, \ldots, \alpha_N^*]^\top$ and $\beta^* = [\beta_0^*, \ldots, \beta_N^*]^\top$ are vectors with entries

$$\alpha_i^* = \mathbb{E}_{q^*(f_i)}[\nabla_{f_i} \log p(y_i \,|\, f_i)|_{f_i = f_i^*}], \tag{20}$$

$$\beta_i^* = \mathbb{E}_{q^*(f_i)}[\nabla_{f_i, f_i} \log p(y_i \,|\, f_i)|_{f_i = f_i^*}]. \tag{21}$$

If we have access to $\alpha$ and $\beta$ then we can determine the posterior's original parameters ($m_f$ and $S_f$). Hence, we follow Adam et al. (2021) and call them dual parameters. Eq. (19) follows because the KL in Eq. (18) has the following form,

$$\mathrm{KL}[q(f) \,\|\, p(f)] = \frac{1}{2}\mathrm{Tr}[K^{-1}S_f] + m_f^\top K^{-1}m_f - \frac{1}{2}\ln|S_f| + const. \tag{22}$$

As such, the derivatives of the KL with respect to the variational parameters ($m_f$ and $S_f$) are given by,

$$\nabla_{m_f}\mathrm{KL}[q(f) \,\|\, p(f)] = K^{-1}m_f, \tag{23}$$

$$\nabla_{S_f}\mathrm{KL}[q(f) \,\|\, p(f)] = \frac{1}{2}[S_f - K^{-1}]. \tag{24}$$

Similarly, the derivatives of the expected log likelihood under the variational posterior are given by,

$$\nabla_{m_f}\mathbb{E}_{q(f_i)}[\log p(y_i \,|\, f_i)] = \mathbb{E}_{q(f_i)}[\nabla_{f_i} \log p(y_i \,|\, f_i)], \tag{25}$$

$$\nabla_{S_f}\mathbb{E}_{q(f_i)}[\log p(y_i \,|\, f_i)] = \frac{1}{2}\mathbb{E}_{q(f_i)}[\nabla_{f_i, f_i} \log p(y_i \,|\, f_i)]. \tag{26}$$

**Dual parameters for MAP objective** The Laplace approximation approximates the posterior using the curvature around the MAP solution $\boldsymbol{f}^*$, which is given by

$$\boldsymbol{f}^* = \arg\max_{\boldsymbol{f}} \mathcal{L}_{\text{LA}}(\boldsymbol{f}) = \arg\max_{\boldsymbol{f}} \sum_{i=1}^{N} \log p(y_i \mid f_i) + \log p(\boldsymbol{f}). \tag{27}$$

Note that the objective does not contain the posterior $q_{\text{LA}}(\boldsymbol{f})$. Taking the derivative of the objective in Eq. (27) w.r.t. $\boldsymbol{f}$ we obtain

$$\nabla_{\boldsymbol{f}} \mathcal{L}_{\text{LA}}(\boldsymbol{f}) = \nabla \log p(\boldsymbol{y} \mid \boldsymbol{f}) - \boldsymbol{K}^{-1} \boldsymbol{f}, \tag{28}$$

$$\nabla^2_{\boldsymbol{f},\boldsymbol{f}} \mathcal{L}_{\text{LA}}(\boldsymbol{f}) = \nabla \log p(\boldsymbol{y} \mid \boldsymbol{f}) - \boldsymbol{K}^{-1}. \tag{29}$$

The Laplace approximation then approximates the posterior with,

$$q_{\text{LA}}(\boldsymbol{f}) = \mathcal{N}(\boldsymbol{f} \mid \boldsymbol{f}^*, \boldsymbol{S}_{\text{LA}}) \quad \text{where} \quad \boldsymbol{S}_{\text{LA}}^{-1} = \nabla^2_{\boldsymbol{f},\boldsymbol{f}} \mathcal{L}_{\text{LA}}(\boldsymbol{f})|_{\boldsymbol{f}=\boldsymbol{f}^*}, \tag{30}$$

where the covariance $\boldsymbol{S}_{\text{LA}}$ is set to the inverse of the Hessian of the objective (in Eq. (27)) at the MAP solution $\boldsymbol{f}^*$. Similar to before, the stationary point of Eq. (27) exhibits a dual parameter decomposition,

$$\boldsymbol{K}\boldsymbol{\alpha}_{\text{LA}}^* = \boldsymbol{f}^*, \qquad \boldsymbol{S}_{\text{LA}}^{-1} = \text{diag}(\boldsymbol{\beta}_{\text{LA}}^*) + \boldsymbol{K}^{-1}, \tag{31}$$

where the dual parameters ($\boldsymbol{\alpha}_{\text{LA}}^*$ and $\boldsymbol{\beta}_{\text{LA}}^*$) are vectors with entries

$$\alpha_{\text{LA},i}^* = \nabla_{f_i} \log p(y_i \mid f_i)|_{f_i=f_i^*}. \tag{32}$$

$$\beta_{\text{LA},i}^* = -\nabla_{f_i,f_i} \log p(y_i \mid f_i)|_{f_i=f_i^*}. \tag{33}$$

Comparing the dual parameters for the ELBO in Eq. (18) to those for the MAP in Eq. (27), we can see that the only difference is the expectation over the posterior.

## A.2 COMPUTATION OF THE DUAL PARAMETERS FOR DIFFERENT LOSSES

As shown in Eq. (8) in the main paper, our method relies on the computation of the dual parameters. We can derive them in general for the family of exponential likelihoods, by rewriting the likelihood in the natural form,

$$\ell(h(f), y) = -\log p(y \mid f) = A(f) - \langle t(y), f \rangle, \tag{34}$$

where $A(f)$ is the log partition function, $h(\cdot)$ is the inverse link function, $t(y)$ are the sufficient statistics which for all our likelihoods is simply $t(y) = y$, and $f$ denotes the NN output before the inverse link function. Then, taking the first and second derivatives of the loss in this form become:

$$\nabla_f \ell(h(f), y_i)|_{f=f_i} = A'(f_i) - y_i \quad \text{and} \quad \nabla^2_{ff} \ell(h(f), y_i)|_{f=f_i} = A''(f_i). \tag{35}$$

In this section, we provide the derivatives needed for the losses used in this work, *i.e.*, the mean squared error (MSE) likelihood, used for the regression tasks, the Bernoulli likelihood, for binary classification, and the categorical likelihood, used for multi-class classification.

For regression, the negative log-likelihood comes from the MSE and for a single input it can be written as

$$\ell(h(f_i), y_i) = \frac{1}{2}(f_i - y_i)^2, \tag{36}$$

where the inverse link function is simply $h(f) = f$. Thus, the first and second derivatives are

$$\nabla_f \ell(\sigma(f), y_i)|_{f=f_i} = f_i - y_i \quad \text{and} \quad \nabla_{ff} \ell(h(f), y_i)|_{f=f_i} = 1. \tag{37}$$

In binary classification, we have a Bernoulli likelihood. The inverse link function is the sigmoid function $h(f) = \sigma(f) = \frac{1}{1+e^{-f}}$, and we can then write the derivatives as follows

$$\nabla_f \ell(h(f), y_i)|_{f=f_i} = \sigma(f_i) - y_i \quad \text{and} \quad \nabla_{ff} \ell(h(f), y_i)|_{f=f_i} = \sigma(f_i)(1 - \sigma(f_i)). \tag{38}$$

In multi-class classification, the function outputs are no longer scalar but a vector of logits $f_{\boldsymbol{w}}(\boldsymbol{x}_i) = \boldsymbol{f}_i \in \mathbb{R}^C$. The inverse link function for the categorical loss is the softmax function $\text{softmax}(\boldsymbol{f}) : \mathbb{R}^C \mapsto [0,1]^C$ defined as $h(\boldsymbol{f}) = \text{softmax}(\boldsymbol{f}) = \frac{\exp(\boldsymbol{f})}{\sum_{c=1}^{C} \exp(\boldsymbol{f}_c)} \in [0,1]^C$. If we write the targets as a one-hot encoded vector $\boldsymbol{y}_i$, we can write the negative log-likelihood as

$$\ell(h(\boldsymbol{f}), \boldsymbol{y}_i) = -\log\left(\boldsymbol{y}_i^\top \text{softmax}(\boldsymbol{f})\right). \tag{39}$$

Finally, we can write the first and second derivatives as follows

$$\begin{aligned}
[\nabla_{\boldsymbol{f}} \ell(h(\boldsymbol{f}), \boldsymbol{y}_i)|_{\boldsymbol{f}=\boldsymbol{f}_i}]_j &= [\text{Softmax}(\boldsymbol{f}) - \boldsymbol{y}_i]_j \\
[\text{diag}\left(\nabla_{\boldsymbol{ff}} \ell(h(\boldsymbol{f}), \boldsymbol{y}_i)|_{\boldsymbol{f}=\boldsymbol{f}_i}\right)]_j &= [\text{Softmax}(\boldsymbol{f})]_j \left(1 - [\text{Softmax}(\boldsymbol{f})]_j\right),
\end{aligned} \tag{40}$$

where we only need the diag given our independence assumption.

## A.3 COMPUTATIONAL COMPLEXITY

In this section, we analyze the computational complexity of SFR. We first need to train a NN $f_{\boldsymbol{w}}$ and then compute the posterior covariance matrix that characterizes our sparse GP representation of the model. The first step then requires the usual complexity cost for training a NN, where defining the cost of a forward pass (and backward pass ) as $[FP]$, this can be roughly sketched as $\mathcal{O}(2EN[FP])$, where $E$ is the number of training epochs and $N$ the number of training samples. Thus, there is no additional cost involved in the training phase of the NN.

**Dual parameter computation** Given the trained network, we construct the covariance function, which is defined through the NTK (Jacot et al., 2018). We relied on the Jacobian contraction formulation implemented using PyTorch (Paszke et al., 2019). Computing the kernel Gram matrix for a batch of $B$ samples with this implementation has a computational complexity of $\mathcal{O}(BC[FP] + B^2 C^2 P)$, where $C$ is the number of outputs and $P$ is the number of parameters of the NN. In Eq. (13), we compute $\boldsymbol{B}_{\boldsymbol{u}}$ iterating through all the $N$ training points to compute all the $\boldsymbol{k}_{\boldsymbol{z}i}$. Then, we use them to compute the outer product, which has a cost of $\mathcal{O}(M^2)$, with $M$ being the number of inducing points. The total cost of this operation becomes $\mathcal{O}(NM^2)$, which is also the dominating cost of our approach.

**Dual updates** The issue with adding new data with standard GP equations is that it's computationally infeasible for large data sets, which are common in deep learning. This is due to the matrix inversion of $(\mathbf{K} + \boldsymbol{\Lambda}^{-1})^{-1}$ which is of complexity $O((N + N_{new})^3)$. There are some linear algebra tricks you can employ that would then make the update $O((N + N_{new})^2)$ but it would still grow with data set size. Incorporating new data into the sparse dual parameters (see Alg. A2) includes the computation of $\boldsymbol{k}_{zi}$ at $N_{new}$ data points which has complexity $\mathcal{O}(N_{new}C[FP] + N_{new}^2 C^2 P)$. Computing the outer products $\boldsymbol{k}_{zi}\hat{\beta}_i\boldsymbol{k}_{zi}$ takes $\mathcal{O}(N_{new}^2 M)$, which is the dominating cost in this step.

## A.4 PRACTICAL CONSIDERATIONS

**Numerical stability** In practical terms, some care needs to be taken during the implementation, both regarding memory consumption and numerical stability. For the former, we batched over the number of training samples, $N$, which keeps the memory consumption in check without the need for any further approximations. The computation of the terms in Eq. (12) requires the inversion of an $M \times M$ matrix, that we implemented in a numerically stable manner using Cholesky factorization. We also introduce a small amount of jitter and clip the GGN matrix for numerical stability where needed (see discussions in App. D). It is also worth noting that we train the neural network in floating point precision (float32) and then switch to double precision (float64) for calculating SFR's dual parameters and making predictions. This is because we ran into issues when using single precision, which we believe was due to the Cholesky decompositions used to implement the matrix inversions. However, we have been testing different approaches for implementing the matrix inversions e.g. using the Woodbury matrix identity and LU and SVD -based direct solves. In some small scale experiments these enabled us to remain in single precision throughout. However, we leave further investigation for future work.

**Hardware** We ran our experiments on a cluster and used a single GPU. The cluster is equipped with four AMD MI250X GPUs based on the 2nd Gen AMD CDNA architecture. A MI250x GPU is a multi-chip module (MCM) with two GPU dies named by AMD Graphics Compute Die (GCD). Each of these dies features 110 compute units (CU) and have access to a 64 GB slice of HBM memory for a total of 220 CUs and 128 GB total memory per MI250x module.

---

**Algorithm A1** Compute SFR's sparse dual parameters

---

**Input:** $\mathcal{D} = \{(\boldsymbol{x}_i, \boldsymbol{y}_i)\}_{i=1}^N$, prior precision $\delta$, number of inducing points $M$, trained NN $f_{\boldsymbol{w}}$, negative log-likelihood function $\ell(\cdot)$, batch size $b$, kernel function $\kappa(\cdot, \cdot)$
**Output:** Duals $\boldsymbol{\alpha}_u, \boldsymbol{B}_u$,
  Sample inducing points $\boldsymbol{Z}$ from $\boldsymbol{X}$
  $\boldsymbol{\alpha_u} = \boldsymbol{0} \in \mathbb{R}^M, \boldsymbol{B_u} = \boldsymbol{0} \in \mathbb{R}^{M \times M}$
  **for** $i$ **in** $[0, \ldots, \text{len}(\mathcal{D})/b]$ **do**
    Select rows indices $\text{SLICE} = (i{\cdot}b : i{\cdot}b + b)$
    $\boldsymbol{X}^{(b)} = \boldsymbol{X}_{\text{SLICE}, :}$
    $\boldsymbol{y}^{(b)} = \boldsymbol{y}_{\text{SLICE}}$
    Compute $\hat{\boldsymbol{\alpha}}^{(b)}, \hat{\boldsymbol{\beta}}^{(b)}$ for $\boldsymbol{X}^{(b)}, \boldsymbol{y}^{(b)}$               ▷ Eq. (11)
    $\boldsymbol{\alpha_u} \leftarrow \boldsymbol{\alpha_u} + \kappa\big(\boldsymbol{Z}, \boldsymbol{X}^{(b)}\big)\, \hat{\boldsymbol{\alpha}}^{(b)}$               ▷ Eq. (13)
    $\boldsymbol{B_u} \leftarrow \boldsymbol{B_u} + \kappa\big(\boldsymbol{Z}, \boldsymbol{X}^{(b)}\big)\, \hat{\boldsymbol{\beta}}^{(b)}\, \kappa\big(\boldsymbol{Z}, \boldsymbol{X}^{(b)}\big)^\top$     ▷ Eq. (13)
  **end for**

---

**Algorithm A2** SFR's dual updates

---

**Input:** Duals $\boldsymbol{\alpha}_u, \boldsymbol{B}_u, \mathcal{D}_{new}\{(\boldsymbol{x}_i, \boldsymbol{y}_i)\}_{i=1}^{N_{new}}$, inducing points $\boldsymbol{Z}$, kernel function $\kappa(\cdot, \cdot)$
**Output:** New duals $\boldsymbol{\alpha}_u^{new}, \boldsymbol{B}_u^{new}$
  **for** $i = 1$ to $N_{new}$ **do**
    Compute $\hat{\alpha}_i, \hat{\beta}_i$ at $\boldsymbol{y}_i$                       ▷ Eq. (11)
    $\boldsymbol{\alpha_u} \leftarrow \boldsymbol{\alpha_u} + \hat{\alpha}_i \kappa(\boldsymbol{Z}, \boldsymbol{x}_i)$                ▷ Eq. (17)
    $\boldsymbol{B_u} \leftarrow \boldsymbol{B_u} + \kappa(\boldsymbol{Z}, \boldsymbol{x}_i) \hat{\beta}_i \kappa(\boldsymbol{Z}, \boldsymbol{x}_i)^\top$       ▷ Eq. (17)
  **end for**

---

### A.5 ALGORITHMS

Alg. A1 details how we compute the sparse dual parameters in Eq. (13). Important considerations include batching over the training data to keep the memory in check. Alg. A2 then details how SFR updates the sparse dual parameters to incorporate information from new data.

## B SFR FOR CONTINUAL LEARNING

This section provides additional details about the related works in the CL literature, the detailed derivation of the SFR-based regularizer described in Sec. 4, and how we extended it for dealing with the multi-output setting in our experiments.

### B.1 RELATED WORK IN THE SCOPE OF CL

Continual learning (CL) hinges on how to deal with a non-stationary training distribution, wherein the training process may overwrite previously learned parameters, causing catastrophic forgetting (Mc-Closkey & Cohen, 1989). CL approaches fall into inference-based, rehearsal-based, and model-based methods (see (Parisi et al., 2019; De Lange et al., 2021) for a detailed survey). The methods in the first category usually tackle this problem with weight-space regularization, such as EWC (Kirkpatrick et al., 2017), SI (Zenke et al., 2017)), or VCL (Nguyen et al., 2018), that induce retention of previously learned information in the weights alone. However, regularizing the weights does not guarantee good quality predictions. Function-space regularization techniques (Li & Hoiem, 2018; Benjamin et al., 2019; Titsias et al., 2020; Buzzega et al., 2020; Pan et al., 2020; Rudner et al., 2022; Achituve et al., 2021) address this by relying on training data subsets for each task to compute a regularization term. These methods can be seen as a hybrid between objective and rehearsal-based methods, since they also store a subset of training data to construct the regularization term. Recent function-based methods, *e.g.*, DER (Buzzega et al., 2020), FROMP (Pan et al., 2020), and S-FSVI (Rudner et al.,

2022), achieve state-of-the-art performance among the objective-based techniques in several CL benchmarks.

## B.2 DERIVATION OF THE REGULARIZER TERM

For CL, the model cannot access the training data from previous tasks up to task $t - 1$, denoted here as $\mathcal{D}_{\text{old}} = \bigcup_{s=1}^{t-1} \mathcal{D}_s$, and for each new task $t$ receives a new chunk of data $\mathcal{D}_{\text{new}} = \mathcal{D}_t$. In contrast, in the supervised learning setting (described in Sec. 2), the NN model has access to all the training data at once, and then the objective we are trying to optimize is

$$\boldsymbol{w}^* = \arg\min_{\boldsymbol{w}} \mathcal{L}(\mathcal{D}_{\text{old}} \cup \mathcal{D}_{\text{new}}, \boldsymbol{w}) = \arg\min_{\boldsymbol{w}} \sum_{i \in \mathcal{D}_{\text{old}} \cup \mathcal{D}_{\text{new}}} \ell(f_{\boldsymbol{w}}(\boldsymbol{x}_i), y_i) + \mathcal{R}(\boldsymbol{w}). \tag{41}$$

Our goal in the continual setting is to replace the missing data coming from $\mathcal{D}_{\text{old}}$ by resorting to a functional regularizer that accounts for that by forcing the predictions of the NNs over the inducing points $\boldsymbol{Z}_t$ to come from $f_{\boldsymbol{w}}^{\text{old}}(\boldsymbol{z}_i) \sim q^{\text{old}}(\boldsymbol{u}), \forall \boldsymbol{z}_i \in \boldsymbol{Z}_t$. As shown in (Khan & Lin, 2017; Adam et al., 2021), we can view the dual parameters derived in Sec. 3 as linked to approximate likelihood terms. Using this insight, we can rewrite the sparse GP posterior for $q(\boldsymbol{u})$ in terms of the following approximate normal distributions:

$$q_{\text{old}}(\boldsymbol{u}) \propto \mathcal{N}(\boldsymbol{u}; \boldsymbol{0}, \boldsymbol{K}_{\boldsymbol{zz}}) \prod_{i \in \mathcal{D}_{\text{old}}} \exp\left(-\frac{\hat{\beta}_i}{2}(\tilde{y}_i - \boldsymbol{a}_i^\top \boldsymbol{u})^2\right) \tag{42}$$

$$\propto p(\boldsymbol{u}) \, \mathcal{N}(\tilde{\boldsymbol{y}}; \boldsymbol{u}, \boldsymbol{B}_{\boldsymbol{u}}), \tag{43}$$

where $\boldsymbol{a}_i^\top = \boldsymbol{k}_{\boldsymbol{z}i}^\top \boldsymbol{K}_{\boldsymbol{zz}}^{-1}$ and $\tilde{y}_i = \hat{\alpha}_i + f_{\boldsymbol{w}^*}(\boldsymbol{x}_i)\hat{\beta}_i$. The sparse GP posterior $q(\boldsymbol{u})$ is proportional to the product between the prior over the inducing points $p(\boldsymbol{u})$ and a Gaussian term $\mathcal{N}(\tilde{\boldsymbol{y}}; \boldsymbol{u}, \boldsymbol{B}_{\boldsymbol{u}})$, which arises from the sparse approximation to the likelihood. Replacing $\boldsymbol{a}_i^\top \boldsymbol{u}$ with $\boldsymbol{x}_i^\top \boldsymbol{w}$, we can see the similarity between the sparse GP and a linear model, for which the kernel is the linear kernel.

Now that we have transformed SFR's dual parameters in to a multivariate form, we can detail the motivation for our function-space CL regularizer and also detail how it differs from previous methods. Using a NN model for function-space VCL (Rudner et al., 2022) has previously been formulated through a converted full GP model for CL. We now show the equivalent for a sparse GP model (Kapoor et al., 2021),

$$q^*(\boldsymbol{u}) = \underset{q(\boldsymbol{u})}{\arg\max} \, \mathcal{L}_{\text{VCL}} = \underset{q(\boldsymbol{u})}{\arg\max} \sum_{i=1}^{N} \mathbb{E}_{q_{\boldsymbol{u}}(f_i)}[\log p(y_i \,|\, f_i)] - \text{KL}[q(\boldsymbol{u}) \,\|\, q_{\text{old}}(\boldsymbol{u})], \tag{44}$$

where $q(\boldsymbol{u})$ is parameterized by $\mathcal{N}(\boldsymbol{m}_{\boldsymbol{u}}, \boldsymbol{S}_{\boldsymbol{u}})$. Substituting Eq. (42) into $\mathcal{L}_{\text{VCL}}$ we get the following expression for sparse VCL,

$$\mathcal{L}_{\text{VCL}} = \sum_{i=1}^{N} \mathbb{E}_{q_{\boldsymbol{u}}(f_i)}[\log p(y_i \,|\, f_i)] + \mathbb{E}_{q_{\boldsymbol{u}}(f_i)}[\log \mathcal{N}(\tilde{\boldsymbol{y}}; \boldsymbol{u}, \boldsymbol{B}_{\boldsymbol{u}})] - \text{KL}[q(\boldsymbol{u}) \,\|\, p(\boldsymbol{u})]. \tag{45}$$

So adding in the expectation of the dual parameters in MVN form is equivalent to doing sparse VCL in function space. Expanding the expectation gives

$$\mathbb{E}_{q(\boldsymbol{u})}[\log \mathcal{N}(\tilde{\boldsymbol{y}} \,|\, \boldsymbol{u}, \boldsymbol{B}_{\boldsymbol{u}})] = -\frac{m}{2} \log(2\pi) - \frac{1}{2} \log |\boldsymbol{B}_{\boldsymbol{u}}|$$
$$- \frac{1}{2}(\tilde{\boldsymbol{y}} - \boldsymbol{m}_{\boldsymbol{u}})^\top [\boldsymbol{B}_{\boldsymbol{u}}]^{-1}(\tilde{\boldsymbol{y}} - \boldsymbol{m}_{\boldsymbol{u}}) - \frac{1}{2}\text{Trace}(\boldsymbol{B}_{\boldsymbol{u}}^{-1} \boldsymbol{S}_{\boldsymbol{u}}). \tag{46}$$

Therefore, we use the quadratic term from $\mathcal{N}(\tilde{\boldsymbol{y}}; \boldsymbol{u}, \boldsymbol{B}_{\boldsymbol{u}})$ as our regularizer in place of the missing data term, the regularizer becomes $(\tilde{\boldsymbol{y}} - \boldsymbol{u})^\top [\boldsymbol{B}_{\boldsymbol{u}}]^{-1}(\tilde{\boldsymbol{y}} - \boldsymbol{u})$. Given that $\tilde{\boldsymbol{y}}$ are in the function space, we can instead use the previous NN predictions $f_{\boldsymbol{w}}^{\text{old}}(\boldsymbol{z}_i)$ and save computations.

Our objective is similar to both Rudner et al. (2022) and Pan et al. (2020) as they both approximate the KL in function-space VCL. However, for scalability we use the sparse function-space VCL formulation, whereas they resort to a subset approach.

### B.3  EXTENDING THE CL REGULARIZER TO MULTI-CLASS SETTINGS

In a multi-class classification setting, the NN can be seen as a function approximator that maps inputs $\boldsymbol{x} \in \mathbb{R}^D$ to the logits $\boldsymbol{f} \in \mathbb{R}^C$, *i.e.*, $f_{\boldsymbol{w}} : \mathbb{R}^D \to \mathbb{R}^C$. As a consequence, the Jacobian matrix is $\boldsymbol{J}_{\boldsymbol{w}}(\boldsymbol{x}) := [\nabla_{\boldsymbol{w}} f_{\boldsymbol{w}}(\boldsymbol{x})]^\top \in \mathbb{R}^{C \times P}$ and the kernel of the associated sparse GP derived in Sec. 3, $\kappa(\boldsymbol{x}, \boldsymbol{x}')$ is a tensor in $\mathbb{R}^{C \times P \times P}$. This means that we need to compute the dual parameters and the matrix $\bar{\boldsymbol{B}}_{t,c}^{-1}$ for all the function's outputs. If we denote $\kappa_c(\boldsymbol{x}, \boldsymbol{x}') \in \mathbb{R}$, as the kernel related to the $c^{\text{th}}$ output function, *i.e.*, the logit for class $c$, we can write the dual parameters as

$$\boldsymbol{\alpha}_{\boldsymbol{u},c} = \sum_{i=1}^N \boldsymbol{k}_{zi,c}\, \hat{\alpha}_{i,c} \in \mathbb{R}^M \quad \text{and} \quad \boldsymbol{B}_{\boldsymbol{u},c} = \sum_{i=1}^N \boldsymbol{k}_{zi,c}\, \hat{\beta}_{i,c}\, \boldsymbol{k}_{zi,c}^\top \in \mathbb{R}^{M \times M}, \quad (47)$$

where $\boldsymbol{k}_{zi,c}$ is the kernel $\kappa_c(\cdot, \cdot)$ computed for the inducing points $\boldsymbol{z}$ in $\boldsymbol{Z}$ and the input point $\boldsymbol{x}_i$ for class $c$. In this setting, the dual variables are

$$\begin{aligned}
\hat{\alpha}_{i,c} &:= [\hat{\boldsymbol{\alpha}}_i]_c = [\nabla_f \log p(y_i \mid f)|_{f=f_i}]_c \in \mathbb{R}, \\
\hat{\beta}_{i,c} &:= \left[\text{diag}(\hat{\boldsymbol{\beta}}_i)\right]_c = \left[-\nabla_{ff}^2 \log p(y_i \mid f)|_{f=f_i}\right]_c \in \mathbb{R},
\end{aligned} \quad (48)$$

where the operator $[\cdot]_c$ is taking the $c^{\text{th}}$ element of a vector. In fact, in the multi-class setting $\hat{\boldsymbol{\alpha}}_i$ is a vector in $\mathbb{R}^C$ and $\hat{\boldsymbol{\beta}}_i$ a matrix in $\mathbb{R}^{C \times C}$. For the SFR-regularizer, at the end of each task, we compute the matrix $\bar{\boldsymbol{B}}_{t,c}^{-1}$ as

$$\bar{\boldsymbol{B}}_{t,c}^{-1} = \boldsymbol{K}_{zz,c}^{-1}\, \boldsymbol{B}_{\boldsymbol{u},c}\, \boldsymbol{K}_{zz,c}^{-1} \in \mathbb{R}^{M \times M}, \quad \text{s.t.} \quad \boldsymbol{B}_{\boldsymbol{u},c} = \sum_{i \in \mathcal{D}_t} \boldsymbol{k}_{zi}\, \hat{\beta}_{i,c}\, \boldsymbol{k}_{zi}^\top. \quad (49)$$

The kernel computation is based on the relationship between the observed data and the parametrized function. For this reason, computing the kernel $\kappa_c$ for classes $c$, for which the model has not observed any data yet, is not meaningful. In practice, for our method we define $\bar{\boldsymbol{B}}_{t,c}^{-1}$ to be $\bar{\boldsymbol{B}}_{t,c}^{-1} = \boldsymbol{I}_M \; \forall c \notin \mathcal{C}_t$, where $\mathcal{C}_t$ is the set of classes observed up to task t. For example, for S-MNIST, after task 1 the set of observed classes for which we will compute the kernel is $\mathcal{C}_1 = \{0, 1\}$, then after task 2 it becomes $\mathcal{C}_1 = \{0, 1, 2, 3\}$, et cetera. Finally, we can rewrite $\mathcal{R}_{\text{SFR}}$ for task $t$ as follows

$$\mathcal{R}_{\text{SFR}}(\boldsymbol{w}, \mathcal{M}_{t-1}) = \frac{1}{2} \sum_{s=1}^{t-1} \sum_{c=1}^{C} \frac{1}{M} \left\| \left[f_{\boldsymbol{w}_s^*}(\boldsymbol{Z}_s)\right]_c - [f_{\boldsymbol{w}}(\boldsymbol{Z}_s)]_c \right\|_{\bar{\boldsymbol{B}}_{s,c}^{-1}}. \quad (50)$$

## C  SFR FOR MODEL-BASED REINFORCEMENT LEARNING

As a final experiment in the main paper (see Sec. 5.2), we provided a demonstration of using SFR in model-based reinforcement learning. As such, this is a direct extension of the uncertainty quantification and representation capabilities already demonstrated in the supervised (and continual) learning examples in the main paper. However, as the RL setting differs from the supervised and CL settings, we present additional background material on it and thus App. C has been separated out to a separate section in the appendix.

In this appendix, we provide a background of the reinforcement learning problem and details of our model-based RL algorithm. See App. D.4 for an overview of the cart pole swing up environment and all details required for reproducing our experiments.

### C.1  RELATED WORK IN THE SCOPE OF MODEL-BASED RL

A key challenge in RL is balancing the trade-off between exploration and exploitation (Sutton & Barto, 2018). One promising direction to balancing this trade-off is to model the uncertainty associated with a learned transition dynamics model and use it to guide exploration. Prior work has taken an expectation over the dynamics model's posterior (Deisenroth & Rasmussen, 2011; Kamthe & Deisenroth, 2018; Chua et al., 2018), sampled from it (akin to Thompson sampling but referred to as posterior sampling in RL) (Dearden et al., 1999; Sasso et al., 2023; Osband et al., 2013), and used it to implement optimism in the face of uncertainty using upper confidence bounds (Curi et al., 2020; Jaksch et al., 2010). No single strategy has emerged as a go-to method and we highlight that these strategies are only as good as their uncertainty estimates. Previous approaches have used GPs (Deisenroth & Rasmussen, 2011; Kamthe & Deisenroth, 2018), ensembles of NNs (Curi et al., 2020; Chua et al., 2018) and variational inference (Gal et al., 2016; Houthooft et al., 2016). However, each method has its pros and cons. In this paper, we present a method which combines some of the pros from NNs with the benefits of GPs.

## C.2 BACKGROUND

We consider environments with states $\boldsymbol{s} \in \mathcal{S} \subseteq \mathbb{R}^{D_s}$, actions $\boldsymbol{a} \in \mathcal{A} \subseteq \mathbb{R}^{D_a}$ and transition dynamics $f : \mathcal{S} \times \mathcal{A} \to \mathcal{S}$, such that $\boldsymbol{s}_{t+1} = f(\boldsymbol{s}_t, \boldsymbol{a}_t) + \boldsymbol{\epsilon}_t$, where $\boldsymbol{\epsilon}_t$ is i.i.d. transition noise. We consider the episodic setting where the system is reset to an initial state $\boldsymbol{s}_0$ at each episode and we assume that there is a known reward function $r : \mathcal{S} \times \mathcal{A} \to \mathbb{R}$. The goal of RL is to find the policy $\pi : \mathcal{S} \to \mathcal{A}$ (from a set of policies $\Pi$) that maximises the sum of discounted rewards in expectation over the transition noise,

$$ J(f, \pi) = \mathbb{E}_{\boldsymbol{\epsilon}_{0:\infty}} \left[ \sum_{t=0}^{\infty} \gamma^t r(\boldsymbol{s}_t, \boldsymbol{a}_t) \right] \quad \text{s.t. } \boldsymbol{s}_{t+1} = f(\boldsymbol{s}_t, \boldsymbol{a}_t) + \boldsymbol{\epsilon}_t, \tag{51} $$

where $\gamma \in [0, 1)$ is a discount factor. In this work, we consider model-based RL where a model of the transition dynamics is learned $f_{\boldsymbol{w}^*} \approx f$ and then used by a planning algorithm. A simple approach is to use the learned dynamics model $f_{\boldsymbol{w}^*}$ and maximise the objective in Eq. (51),

$$ \pi^{\text{Greedy}} = \arg\max_{\pi \in \Pi} J(f_{\boldsymbol{w}^*}, \pi). \tag{52} $$

However, we can leverage the method in Sec. 2 to obtain a function-space posterior over the learned dynamics $q_{\boldsymbol{u}}(f \mid \mathcal{D})$, where $\mathcal{D}$ represents the state transition data set $\mathcal{D} = \{(s_i, a_i), s_{i+1}\}_{i=0}^{N}$. Importantly, the uncertainty represented by this posterior distribution can be used to balance the exploration-exploitation trade-off, using approaches such as posterior sampling (Osband & Van Roy, 2017; Osband et al., 2013),

$$ \pi^{\text{PS}} = \arg\max_{\pi \in \Pi} J(\tilde{f}, \pi) \quad \text{s.t. } \tilde{f} \sim q_{\boldsymbol{u}}(f \mid \mathcal{D}), \tag{53} $$

where a function $\tilde{f}$ is sampled from the (approximate) posterior $q_{\boldsymbol{u}}(f \mid \mathcal{D})$ and used to find a policy as in Eq. (52). Intuitively, this strategy will explore where the model has high uncertainty, which in turn will reduce the model's uncertainty as data is collected and used to train the model. This strategy should perform well in environments which are hard to explore, for example, those with sparse rewards.

## C.3 ALGORITHM

In practice, it is common to approximate the infinite sum in Eq. (53) by planning over a finite horizon and bootstrapping with a learned value function. Following this approach, our planning algorithim is given by,

$$ \pi^{\text{PS}}(\boldsymbol{s}) = \arg\max_{\boldsymbol{a}_0} \max_{\boldsymbol{a}_{1:H}} \mathbb{E}\left[ \sum_{t=0}^{H-1} \gamma^t r(\boldsymbol{s}_t, \boldsymbol{a}_t) \mid \boldsymbol{s}_0 = \boldsymbol{s} \right] + Q_\theta(\boldsymbol{s}_H, \boldsymbol{a}_H), \tag{54} $$

such that $\boldsymbol{s}_{t+1} = \tilde{f}(\boldsymbol{s}_t, \boldsymbol{a}_t) + \boldsymbol{\epsilon}_t$, where $\tilde{f} \sim q_{\boldsymbol{u}}(f \mid \mathcal{D})$ is a sample from the dynamics posterior and $Q_\theta(\boldsymbol{s}, \boldsymbol{a}) \approx \mathbb{E}\left[ \sum_{t=0}^{H-1} \gamma^t r(\boldsymbol{s}_t, \boldsymbol{a}_t) \mid \boldsymbol{s}_0 = \boldsymbol{s}, \boldsymbol{a}_0 = \boldsymbol{a} \right]$ is a learned value function with parameters $\theta$. We use deep deterministic policy gradient (DDPG, Lillicrap et al., 2016) to learn the action value function $Q_\theta$. Note that we also learn a policy but its sole purpose is for learning the value function.

**Model predictive path integral (MPPI)** We adopt model predictive path integral (MPPI) control (Pan et al., 2015; Williams et al., 2017) to plan online over a $H$ step horizon and then bootstrap with a learned value function. MPPI is an online planning algorithim which iteratively improves the action trajectory $\boldsymbol{a}_{t:t+H}$ using samples. At each iteration $j$, $N$ trajectories are sampled according to the currect action trajectory $\boldsymbol{a}_{t:t+H}^j$. The top $K$ trajectories with highest returns $\sum_{h=0}^{H} = \gamma^h r(\boldsymbol{s}_{t+h}^j, \boldsymbol{a}_{t+h}^j) + \gamma Q_\theta(\boldsymbol{s}_H, \boldsymbol{a}_H)$ are selected. The next action trajectory $\boldsymbol{a}_{t:t+H}^{j+1}$ is then computed by taking the weighted average of the top $K$ trajectories with weights from the softmax over returns from top $K$ trajectories. After $J$ iterations the first action is executed in the environment.

See App. D.4 for details of our experiments and results.

## D EXPERIMENT DETAILS

In this section, we provide further details for the experiments presented in the main paper.

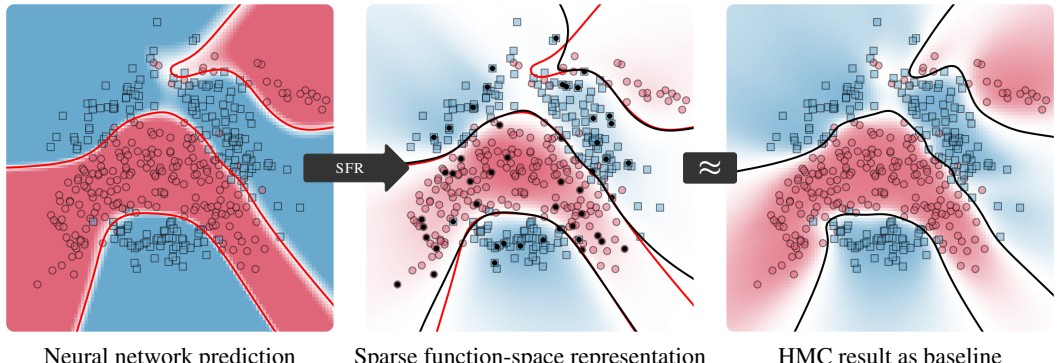

| Neural network prediction | Sparse function-space representation | HMC result as baseline |

Figure A5: **Uncertainty quantification** for binary classification (■ vs. ●). We convert the trained neural network (left) to a sparse GP model that summarizes all data onto a sparse set of inducing points ● (middle). This gives similar behaviour as running full Hamiltonian Monte Carlo (HMC) on the original neural network model weights (right). Marginal uncertainty depicted by colour intensity.

**Table bolding**  In tables throughout the paper, we mark the best-performing method/setting in bold. Following this, we check with a two-tailed $t$-test whether the bolded entry significantly differs from the rest. Those not differing from the best performing one with a significance level of 5% are also bolded.

**Toy example**  The illustrative examples in Figs. 1 and A5 include toy problems for regression and classification. The 1D regression problem (Fig. 1) uses an MLP with two hidden layers (64 hidden units each, with tanh activation), whereas the 2D banana binary classification problem uses an MLP with two hidden layers (64 hidden units each, with sigmoid activation). Both examples use Adam as the optimizer. We include Python notebooks for trying out the models. The illustrative HMC baseline in Fig. A5 (right) was implemented with the help of hamiltorch[2]. We used ten chains with 10,000 samples each and remove 2000 samples as burn-in. The HMC result is more expressive but the results are still similar in terms of quantifying the uncertainty in low-data regions.

## D.1  SUPERVISED LEARNING EXPERIMENTS

In this section, we provide details of the supervised learning experiments reported in the main table. In particular, we discuss our choice of reporting the results without any prior precision tuning in App. D.1.1, the details of the UCI data set experiments in App. D.1.2 and the ones for image data sets in App. D.1.3.

### D.1.1  PRIOR PRECISION TUNING

It is common practice to tune the prior precision $\delta$ after training the NN. This *post hoc* step is an additional tuning to find a prior precision $\delta_{\text{tune}}$, which has a better NLPD on the validation set than the $\delta$ used to train the NN. Note that this leads to a different prior precision being used for training and inference. We highlight that this technically invalidates the Laplace/SFR methods, as the NN weights are no longer the MAP weights. Nevertheless, in the following sections, App. D.1.2 and App. D.1.3, we report results both when the prior precision $\delta$ is tuned and when it is not. When tuning the prior precision $\delta$, SFR matches the performance of the Laplace methods (BNN/GLM). In contrast to the other methods, SFR already performs well without tuning the prior precision $\delta$, whereas the Laplace methods require this additional *post hoc* optimization step to achieve good results.

### D.1.2  UCI EXPERIMENT CONFIGURATION

In Sec. 5.1 in the main paper, we showed experiments on 8 UCI data sets (Dua & Graff, 2017). We used a two-layer MLP with width 50, tanh activation functions and a

---

[2]https://github.com/AdamCobb/hamiltorch

Table A4: **UCI classification with prior precision $\delta$ tuning** Comparisons on UCI data with negative log predictive density (NLPD$\pm$std, lower better). SFR, with $M = 20\%$ of $N$, is on par with the Laplace approximation (BNN/GLM) and outperforms the GP subset when the prior precision ($\delta$) is tuned. SFR-NN uses the NN as its posterior mean function. GP subset, GP predictive, and SFR use the same inducing points.

| | $N$ | $D$ | $C$ | NN MAP | LAPLACE full | LAPLACE GLM full | GP predictive $M = 20\%$ of $N$ | GP subset $M = 20\%$ of $N$ | SFR $M = 20\%$ of $N$ | SFR-NN $M = 20\%$ of $N$ |
|---|---|---|---|---|---|---|---|---|---|---|
| AUSTRALIAN | 690 | 14 | 2 | 0.33$\pm$0.08 | **0.33**$\pm$0.05 | **0.33**$\pm$0.06 | **0.42**$\pm$0.15 | 0.43$\pm$0.04 | **0.33**$\pm$0.07 | **0.33**$\pm$0.07 |
| BREAST CANCER | 683 | 10 | 2 | 0.10$\pm$0.04 | **0.10**$\pm$0.04 | **0.10**$\pm$0.04 | **0.10**$\pm$0.04 | 0.12$\pm$0.03 | **0.10**$\pm$0.03 | **0.10**$\pm$0.04 |
| DIGITS | 351 | 34 | 2 | 0.09$\pm$0.01 | 0.14$\pm$0.02 | **0.10**$\pm$0.01 | **0.09**$\pm$0.01 | 0.22$\pm$0.09 | 0.11$\pm$0.01 | **0.09**$\pm$0.01 |
| GLASS | 214 | 9 | 6 | 0.94$\pm$0.16 | **0.92**$\pm$0.15 | **0.92**$\pm$0.14 | **0.92**$\pm$0.16 | 1.18$\pm$0.16 | **0.92**$\pm$0.05 | 0.93$\pm$0.17 |
| IONOSPHERE | 846 | 18 | 4 | 0.33$\pm$0.06 | 0.36$\pm$0.04 | **0.32**$\pm$0.05 | 0.34$\pm$0.06 | 0.41$\pm$0.05 | **0.32**$\pm$0.07 | 0.33$\pm$0.06 |
| SATELLITE | 1000 | 21 | 3 | 0.27$\pm$0.02 | 0.30$\pm$0.01 | 0.27$\pm$0.02 | 0.51$\pm$0.57 | 0.34$\pm$0.01 | **0.26**$\pm$0.02 | 0.27$\pm$0.02 |
| VEHICLE | 1797 | 64 | 10 | 0.35$\pm$0.05 | 0.37$\pm$0.04 | 0.35$\pm$0.05 | 0.34$\pm$0.05 | 0.46$\pm$0.04 | 0.35$\pm$0.05 | 0.34$\pm$0.05 |
| WAVEFORM | 6435 | 35 | 6 | 0.33$\pm$0.01 | **0.33**$\pm$0.01 | **0.33**$\pm$0.01 | **0.33**$\pm$0.01 | 0.40$\pm$0.05 | **0.32**$\pm$0.02 | **0.33**$\pm$0.01 |

Table A5: **UCI classification without prior precision $\delta$ tuning:** Comparisons on UCI data with negative log predictive density (NLPD$\pm$std, lower better). SFR, with $M = 20\%$ of $N$, outperforms the Laplace approximation (BNN/GLM) (Daxberger et al., 2021), GP predictive from Immer et al. (2021b) and the GP subset when the prior precision ($\delta$) is not tuned. SFR-NN uses the NN as the posterior mean function. GP subset, GP predictive and, SFR use the same inducing points.

| | $N$ | $D$ | $C$ | NN MAP | LAPLACE full | LAPLACE GLM full | GP predictive $M = 20\%$ of $N$ | GP subset $M = 20\%$ of $N$ | SFR $M = 20\%$ of $N$ | SFR-NN $M = 20\%$ of $N$ |
|---|---|---|---|---|---|---|---|---|---|---|
| AUSTRALIAN | 690 | 14 | 2 | 0.33$\pm$0.08 | 0.71$\pm$0.02 | 0.41$\pm$0.03 | **0.34**$\pm$0.05 | 0.42$\pm$0.05 | **0.33**$\pm$0.07 | **0.33**$\pm$0.06 |
| BREAST CANCER | 683 | 10 | 2 | 0.10$\pm$0.04 | 0.73$\pm$0.06 | 0.36$\pm$0.05 | 0.22$\pm$0.01 | 0.20$\pm$0.07 | **0.16**$\pm$0.02 | **0.16**$\pm$0.02 |
| DIGITS | 351 | 34 | 2 | 0.09$\pm$0.01 | 2.34$\pm$0.02 | 3.12$\pm$0.22 | 1.09$\pm$0.03 | 1.10$\pm$0.06 | **1.07**$\pm$0.05 | 1.06$\pm$0.06 |
| GLASS | 214 | 9 | 6 | 0.94$\pm$0.16 | 1.79$\pm$0.03 | 1.65$\pm$0.19 | 1.12$\pm$0.09 | 1.21$\pm$0.15 | **0.86**$\pm$0.11 | 0.97$\pm$0.10 |
| IONOSPHERE | 846 | 18 | 4 | 0.33$\pm$0.06 | 0.70$\pm$0.02 | **0.34**$\pm$0.04 | 0.34$\pm$0.05 | 0.43$\pm$0.04 | **0.32**$\pm$0.06 | 0.33$\pm$0.05 |
| SATELLITE | 1000 | 21 | 3 | 0.27$\pm$0.02 | 1.80$\pm$0.02 | 0.80$\pm$0.03 | 0.29$\pm$0.01 | 0.34$\pm$0.01 | **0.26**$\pm$0.02 | 0.27$\pm$0.02 |
| VEHICLE | 1797 | 64 | 10 | 0.35$\pm$0.05 | 1.39$\pm$0.02 | 1.52$\pm$0.02 | 0.77$\pm$0.03 | 0.78$\pm$0.03 | **0.72**$\pm$0.04 | 0.76$\pm$0.03 |
| WAVEFORM | 6435 | 35 | 6 | 0.33$\pm$0.01 | 1.10$\pm$0.01 | 1.03$\pm$0.03 | 0.62$\pm$0.02 | 0.62$\pm$0.05 | **0.52**$\pm$0.05 | 0.55$\pm$0.03 |

70% (train) : 15% (validation) : 15% (test) data split. We trained the NN using Adam (Kingma & Ba, 2015) with a learning rate of $10^{-4}$ and a batch size of 128. Training was stopped when the validation loss stopped decreasing after 1000 steps. The checkpoint with the lowest validation loss was used as the NN MAP. Each experiment was run for 5 seeds. For the HouseElectric experiment in Sec. 5.1 of the main paper, we keep the same 70% (train) : 15% (validation) : 15% (test) data split, but we switched to a deeper and wider MLP with 3 layers and 512 units per layer, with $\tanh$ activation functions. The MLP is trained with a learning rate of $10^{-3}$ for 100 epochs and a batch size of 128. To adhere with the number of inducing points reported in the paper of Wang et al. (2019), we used $M = 1024$. We ran the experiment for five random seeds.

### D.1.3 IMAGE EXPERIMENT CONFIGURATION

Sec. 5.1 in the main paper contains the experiments on two image data sets, *i.e.*, FMNIST (Xiao et al., 2017) and CIFAR-10 (Krizhevsky et al., 2009). For all the experiments, we used a CNN composed of three convolutional layers with ReLU activations, followed by three linear layers with $\tanh$ activation functions. This model follows the implementation standardized in Schneider et al. (2019) without using dropout layers, as also done in Immer et al. (2021b). We trained the NNs using Adam (Kingma & Ba, 2015) for 500 epochs with a batch size of 128. For FMNIST we used a learning rate of $10^{-4}$ and for CIFAR-10 we increased it to $10^{-3}$. The training was stopped when the validation loss stopped decreasing after 15 steps. We performed a grid search to select the best value of the prior precision. We reported the runs performed with $\delta = 0.0018$ for CIFAR-10 and $\delta = 0.0013$ for F-MNIST. The runs are repeated five times with different random seeds. As discussed in Sec. 5.1 and in more detail in App. D.1.1, we reported only the results obtained without any *post hoc* tuning for the prior precision in the main paper. However, here in Table A6, we also report the results obtained by tuning the prior precision for SFR, GP subset and Laplace GLM/BNN. For SFR and GP subset, we implemented the $\delta$ tuning as a BO with 20 trials, while we used the built-in methods provided by the Laplace Redux library (Daxberger et al., 2021), with which we implemented the Laplace baselines.

The improvement in performance for the Laplace methods is notable for both FMNIST and CIFAR-10. SFR matches their performance after tuning, with the advantage of performing better than them without the need of *post hoc* prior precision tuning. It is also important to note that in Table A6, we

Table A6: **SFR results on image data sets** Classification results on image data using CNN. We report NLPD, accuracy, expected calibration error (ECE) and AUROC (mean±std over 5 seeds). SFR outperforms the GP SUBSET method and the LAPLACE methods without any $\delta$ *post hoc* tuning, and matches them after the *post hoc* prior precision tuning. We report the results for SFR using different $M$ values, with and without a *post hoc* tuning for $\delta$.

| | | MODEL | $M$ | NLPD ↓ | ACC. (%) ↑ | ECE ↓ | AUROC ↑ |
|---|---|---|---|---|---|---|---|
| **F-MNIST** | | NN MAP | - | 0.23±0.01 | 91.98±0.44 | 0.01±0.00 | 0.83±0.05 |
| | | LAPLACE DIAG | - | 2.42±0.02 | 10.21±0.66 | 0.10±0.02 | 0.50±0.03 |
| | | LAPLACE KRON | - | 2.39±0.01 | 9.87±0.66 | 0.07±0.01 | 0.51±0.02 |
| | | LAPLACE GLM DIAG | - | 1.66±0.02 | 65.19±2.21 | 0.44±0.02 | 0.67±0.03 |
| | | LAPLACE GLM KRON | - | 1.09±0.04 | 84.79±1.96 | 0.47±0.01 | 0.96±0.01 |
| | | GP PREDICTIVE | 3200 | 0.47±0.06 | 91.51±0.45 | 0.23±0.03 | **0.97**±**0.01** |
| | | GP SUBSET | 512 | 1.83±0.68 | 48.48±14.93 | 0.26±0.08 | 0.77±0.10 |
| | | | 1024 | 1.31±0.30 | 66.90±14.32 | 0.33±0.06 | 0.88±0.04 |
| | | | 2048 | 0.97±0.13 | 77.32±8.83 | 0.30±0.08 | 0.93±0.03 |
| | | | 3200 | 0.79±0.09 | 82.52±4.10 | 0.29±0.05 | **0.95**±**0.02** |
| | | SFR (Ours) | 512 | 0.38±0.02 | 90.99±0.55 | 0.15±0.01 | **0.95**±**0.02** |
| | | | 1024 | 0.34±0.03 | 91.41±0.46 | 0.12±0.02 | **0.95**±**0.02** |
| | | | 2048 | 0.30±0.01 | **91.68**±**0.51** | 0.09±0.00 | 0.95±0.02 |
| | | | 3200 | 0.29±0.02 | **91.74**±**0.47** | 0.08±0.01 | **0.96**±**0.01** |
| | | SFR-NN (Ours) | 512 | 0.37±0.03 | 91.31±0.64 | 0.15±0.01 | **0.97**±**0.02** |
| | | | 3200 | 0.31±0.03 | **91.86**±**0.40** | 0.10±0.02 | **0.97**±**0.01** |
| | **δ tuning** | LAPLACE DIAG | - | 1.88±0.01 | 41.02±4.90 | 0.22±0.06 | 0.64±0.04 |
| | | LAPLACE KRON | - | 1.39±0.01 | 82.31±3.81 | 0.55±0.04 | 0.92±0.05 |
| | | LAPLACE GLM DIAG | - | 0.52±0.02 | 90.97±0.51 | 0.26±0.01 | **0.96**±**0.02** |
| | | LAPLACE GLM KRON | - | 0.25±0.01 | **91.78**±**0.54** | 0.05±0.00 | 0.91±0.04 |
| | | GP PREDICTIVE | 3200 | **0.23**±**0.01** | **92.09**±**0.20** | **0.01**±**0.00** | 0.87±0.02 |
| | | GP SUBSET | 512 | 1.61±0.20 | 50.26±12.36 | 0.22±0.09 | 0.80±0.07 |
| | | | 1024 | 1.30±0.27 | 65.70±16.14 | 0.30±0.14 | **0.82**±**0.15** |
| | | | 2048 | 0.96±0.13 | 77.49±8.65 | 0.31±0.07 | 0.93±0.03 |
| | | | 3200 | 0.79±0.09 | 82.56±3.79 | 0.29±0.05 | **0.95**±**0.02** |
| | | SFR (Ours) | 512 | 0.37±0.01 | 89.69±1.53 | 0.11±0.03 | **0.88**±**0.09** |
| | | | 1024 | 0.34±0.03 | 91.38±0.55 | 0.12±0.02 | **0.95**±**0.02** |
| | | | 2048 | 0.30±0.01 | 91.57±0.44 | 0.09±0.00 | 0.95±0.02 |
| | | | 3200 | 0.29±0.02 | **91.82**±**0.50** | 0.08±0.01 | 0.96±0.01 |
| | | SFR-NN (Ours) | 512 | **0.23**±**0.01** | 91.87±0.55 | **0.01**±**0.00** | 0.85±0.05 |
| | | | 3200 | **0.23**±**0.01** | **92.14**±**0.22** | **0.01**±**0.00** | 0.86±0.02 |
| **CIFAR-10** | | NN MAP | - | 0.69±0.03 | 77.00±1.04 | 0.04±0.01 | 0.85±0.02 |
| | | LAPLACE DIAG | - | 2.37±0.05 | 10.08±0.24 | 0.06±0.02 | 0.48±0.01 |
| | | LAPLACE KRON | - | 2.36±0.01 | 9.78±0.41 | 0.06±0.01 | 0.49±0.01 |
| | | LAPLACE GLM DIAG | - | 1.33±0.05 | 71.96±1.38 | 0.39±0.02 | **0.82**±**0.03** |
| | | LAPLACE GLM KRON | - | 1.04±0.08 | 75.56±1.63 | 0.30±0.03 | 0.64±0.04 |
| | | GP PREDICTIVE | 3200 | 0.90±0.02 | 76.07±1.17 | 0.23±0.02 | 0.79±0.02 |
| | | GP SUBSET | 512 | 1.56±0.08 | 50.89±5.30 | 0.20±0.05 | 0.64±0.06 |
| | | | 1024 | 1.38±0.08 | 58.30±6.35 | 0.22±0.07 | 0.64±0.10 |
| | | | 2048 | 1.18±0.06 | 66.40±3.69 | 0.24±0.04 | 0.71±0.05 |
| | | | 3200 | 1.08±0.05 | 69.75±3.23 | 0.24±0.03 | 0.75±0.01 |
| | | SFR (Ours) | 512 | 0.86±0.02 | **77.58**±**0.95** | 0.24±0.01 | 0.80±0.02 |
| | | | 1024 | 0.79±0.02 | **78.39**±**0.89** | 0.20±0.01 | 0.80±0.02 |
| | | | 2048 | 0.74±0.02 | **78.40**±**0.83** | 0.16±0.00 | 0.79±0.02 |
| | | | 3200 | 0.72±0.02 | **78.48**±**0.98** | 0.14±0.01 | 0.79±0.02 |
| | | SFR-NN (Ours) | 512 | 0.89±0.02 | 75.97±1.40 | 0.22±0.02 | 0.79±0.02 |
| | | | 3200 | 0.79±0.02 | 76.59±1.21 | 0.16±0.02 | 0.79±0.03 |
| | **δ tuning** | LAPLACE DIAG | - | 2.13±0.01 | 28.89±4.80 | 0.14±0.05 | 0.39±0.06 |
| | | LAPLACE KRON | - | 2.13±0.01 | 28.60±3.03 | 0.14±0.03 | 0.35±0.06 |
| | | LAPLACE GLM DIAG | - | 0.74±0.04 | **76.90**±**1.25** | 0.11±0.02 | **0.81**±**0.03** |
| | | LAPLACE GLM KRON | - | **0.69**±**0.03** | **76.95**±**1.12** | 0.05±0.02 | **0.82**±**0.03** |
| | | GP PREDICTIVE | 3200 | **0.68**±**0.03** | 76.94±1.03 | **0.02**±**0.01** | **0.84**±**0.02** |
| | | GP SUBSET | 512 | 1.56±0.08 | 50.62±5.51 | 0.20±0.05 | 0.64±0.06 |
| | | | 1024 | 1.37±0.08 | 57.95±6.21 | 0.22±0.06 | 0.65±0.10 |
| | | | 2048 | 1.18±0.06 | 66.24±3.83 | 0.24±0.04 | 0.71±0.05 |
| | | | 3200 | 1.08±0.05 | 69.88±2.76 | 0.24±0.03 | 0.75±0.01 |
| | | SFR (Ours) | 512 | 0.78±0.06 | 75.12±0.93 | 0.11±0.06 | **0.82**±**0.02** |
| | | | 1024 | 0.74±0.05 | **77.19**±**1.12** | 0.12±0.06 | **0.82**±**0.03** |
| | | | 2048 | **0.71**±**0.03** | **77.48**±**1.10** | 0.11±0.05 | **0.82**±**0.03** |
| | | | 3200 | **0.70**±**0.03** | **77.67**±**1.01** | 0.10±0.04 | **0.82**±**0.03** |
| | | SFR-NN (Ours) | 512 | **0.68**±**0.03** | 76.92±1.15 | **0.02**±**0.01** | **0.84**±**0.02** |
| | | | 3200 | **0.68**±**0.03** | 76.94±1.16 | **0.02**±**0.01** | **0.84**±**0.02** |

report the expected calibration error (ECE) metric, which is problematic as a standalone metric and should be carefully assessed considering the accuracy. For instance, in the CIFAR-10 rows without $\delta$

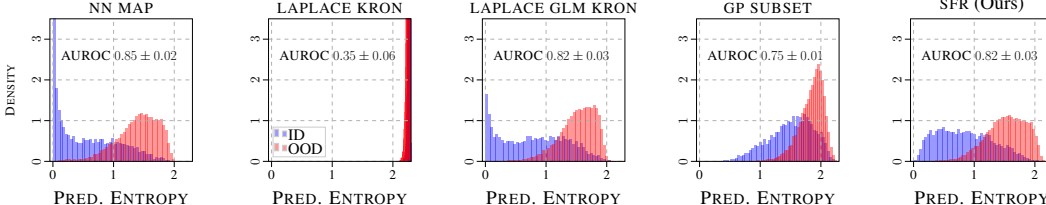

Figure A6: **OOD detection with CNN:** Histograms showing each method's predictive entropy at in-distribution (ID) data (CIFAR-10 test set, blue) where lower is better and at out-of-distribution (OOD) data (SVHN test set, red) where higher is better. The prior precision $\delta$ was tuned *post hoc*.

tuning, the ECE is very low for Laplace DIAG and KRON even if the accuracy is very poor, meaning that the model is well calibrated but producing inaccurate predictions.

Table A6 results show how SFR consistently outperforms the GP subset method, with both lower and higher numbers $M$ of inducing points. This shows that our dual parameterization better captures the information from all training data.

### D.1.4    OUT-OF-DISTRIBUTION DETECTION EXPERIMENTS

In Sec. 5.1 we evaluated SFR's OOD detection. In all experiments, we tuned the prior precision *post hoc*. Fig. A6 provides further OOD detection results when training on CIFAR-10 and using SVHN (Netzer et al., 2011) as the OOD data set.

### D.2    CONTINUAL LEARNING EXPERIMENT DETAILS

In Sec. 5.2 in the main paper, we showed how the regularizer described in Sec. 4 can be used in CL to retain a compact representation of the NN. We conducted our experiments on three common CL data sets (De Lange et al., 2021; Pan et al., 2020; Rudner et al., 2022):

- Split-MNIST (S-MNIST), that consists of five binary classification tasks, each on a pair of MNIST classes, *i.e.*, ⓪ and ①, ② and ③, . . . , ⑧ and ⑨.
- Split-FashionMNIST (S-FMNIST), composed of five binary classification tasks, similar to S-MNIST, but using Fashion-MNIST.
- Permuted-MNIST (P-MNIST), that consists of ten tasks, where each one is a ten-way classification task on the entire MNIST data set, where the images' pixels are randomly permuted.

Following Pan et al. (2020); Rudner et al. (2022), we used a two-layer MLP with 256 hidden units and ReLU activations for S-MNIST/S-FMNIST and a two-layer MLP with 100 hidden units and ReLU activations for P-MNIST, to ensure consistency with the other methods. In our experiments we only consider the single-head (SH) setting. This means that for S-MNIST and S-FMNIST, we are dealing with a *Class-Incremental Learning* (CIL) scenario, where for each task a pair of disjoint classes are presented to the model. The P-MNIST experiments fall into the *Domain-Incremental Learning* (DIL) setting, where the model deals with a data-distribution shift, by progressively observing tasks with different permutations of the image pixels that need to be classified into ten different classes.

In our experiments, we compared the performance of the SFR-based regularizer against two families of methods: weight-space regularization methods and function-space regularization ones. For the weight-space methods, we considered Online-EWC (Schwarz et al., 2018), SI (Zenke et al., 2017), and VCL (Nguyen et al., 2018). For the function-space methods, we considered DER (Buzzega et al., 2020), FROMP (Pan et al., 2020), and S-FSVI (Rudner et al., 2022).

For our method, DER, Online-EWC, and SI, we relied on the Mammoth framework proposed by Buzzega et al. (2020). We used the available open-source FROMP codebase to conduct the experiments with FROMP, and the S-FSVI codebase to run the experiments on S-FSVI and VCL. We selected the hyperparameters to use with all these methods by trying to adapt the ones pointed out in the original papers and reference implementations. For this reason, we relied on SGD for DER,

Online-EWC, and SI, which requires higher learning rates, compared to the others methods for which we used Adam (Kingma & Ba, 2015), *i.e.*, our SFR regularizer, FROMP, S-FSVI, and VCL.

For the methods relying on a stored subset of training data for each task, we make sure that the number of data points stored are the same for each method. For DER, which does not select the rehearsal training samples at task boundaries, we cannot directly apply the definition of points per task ($M$), and thus we set the dimension of the reservoir buffer to be $T \cdot M$, *i.e.*, total number of tasks times the points per task.

The results reported in Table 2 are obtained using the following hyperparameters and repeating the runs five times with different seeds. The results are in-line with what has been reported in previous work, notably even better for some of the baseline methods. Thus, we consider the presented results to give a realistic picture of the relative performance of the different methods, demonstrating a clear benefit of using SFR for CL.

**S-MNIST (40 pts./task):** **Online-EWC**: SGD, learning rate 0.03, batch size 10, number of epochs 1, $\lambda = 90$, $\gamma = 1.0$. **SI**: SGD, learning rate 0.1, batch size 10, number of epochs 1, $c = 1.0$, $\xi = 0.9$. **VCL (with coresets)**: Adam, learning rate 0.001, batch size 256, number of epochs 100, selection method for coresets random choice. **DER**: SGD with learning rate 0.03, batch size 10, buffer batch size 10, number epochs 1, $\alpha = 0.3$, reservoir buffer size 200. **FROMP**: Adam, learning rate $1 \cdot 10^{-4}$, batch size 32, number of epochs 10, $\tau = 10$, random selection for memorable points. **S-FSVI**: Adam, learning rate $5 \cdot 10^{-4}$, batch size 128, number of epochs 10. **SFR**: Adam, learning rate $3 \cdot 10^{-4}$, batch size 32, $\tau = 1$, $\delta = 0.0001$, number of epochs 1.

**S-MNIST (200 pts./task):** **Online-EWC**: SGD, learning rate 0.03, batch size 10, number of epochs 1, $\lambda = 90$, $\gamma = 1.0$. **SI**: SGD, learning rate 0.1, batch size 10, number of epochs 1, $c = 1.0$, $\xi = 0.9$. **VCL (with coresets)**: Adam, learning rate 0.001, batch size 256, number of epochs 100, selection method for coresets random choice. **DER**: SGD, learning rate 0.03, batch size 10, buffer batch size 10, number epochs 1, $\alpha$ 0.2, reservoir buffer size 1000. **FROMP**: Adam, learning rate $1 \cdot 10^{-4}$, batch size 32, number of epochs 10, $\tau = 10$, random selection for memorable points. **S-FSVI**: Adam, learning rate $5 \cdot 10^{-4}$, batch size 128, number of epochs 80. **SFR**: Adam, learning rate $1 \cdot 10^{-4}$, batch size 32, $\tau = 0.5$, $\delta = 1 \cdot 10^{-5}$, number of epochs 5.

**S-FMNIST (200 pts./task):** **Online-EWC**: SGD, learning rate 0.03, batch size 10, number of epochs 5, $\lambda = 90$, $\gamma = 1.0$. **SI**: SGD, learning rate 0.1, batch size 10, number of epochs 5, $c = 1.0$, $\xi = 0.9$. **VCL (with coresets)**: Adam, learning rate $1 \cdot 10^{-3}$, batch size 256, number of epochs 100, selection method for coresets random choice. **DER**: SGD, learning rate 0.03, batch size 10, buffer batch size 10, number epochs 5, $\alpha = 0.2$, reservoir buffer size 200. **FROMP**: Adam, learning rate $1 \cdot 10^{-4}$, batch size 32, number of epochs 10, $\tau = 10$, random selection for memorable points. **S-FSVI**: Adam, learning rate $5 \cdot 10^{-4}$, batch size 128, number of epochs 60. **SFR**: Adam, learning rate $3 \cdot 10^{-4}$, batch size 32, $\tau = 1.0$, $\delta = 0.0001$, number of epochs 5.

**P-MNIST (200 pts./task):** **Online-EWC**: SGD, learning rate 0.1, batch size 128, number of epochs 10, $\lambda = 0.7$, $\gamma = 1.0$. **SI**: SGD, learning rate 0.1, batch size 138, number of epochs 10, $c = 0.5$, $\xi = 1.0$. **VCL (with coresets)**: Adam, learning rate 0.001, batch size 256, number of epochs 100, selection method for coresets random choice. **DER**: SGD, learning rate 0.01, batch size 128, buffer batch size 128, number epochs 10, $\alpha = 0.3$, reservoir buffer size 2000. **FROMP**: Adam, learning rate $1 \cdot 10^{-3}$, batch size 128, number of epochs 10, $\tau = 0.5$, lambda descend selection for memorable points. **S-FSVI**: Adam, learning rate $5 \cdot 10^{-4}$, batch size 128, number of epochs 10. **SFR**: Adam, learning rate $3 \cdot 10^{-4}$, batch size 64, $\tau = 1.0$, $\delta = 0.0001$.

Note that we used the same hyperparameters for the SFR ablation experiments.

### D.3 INCORPORATING NEW DATA VIA DUAL UPDATES EXPERIMENT DETAILS

This section details how we configured and ran our incorporating new data via dual conditioning experiments.

In all experiments, we used a two-layer MLP with 128 hidden units and $\tanh$ activations. We trained the NN using Adam (Kingma & Ba, 2015) with a learning rate of $10^{-4}$ and a batch size of $50$. Training was stopped when the validation loss stopped decreasing after 100 steps. The checkpoint

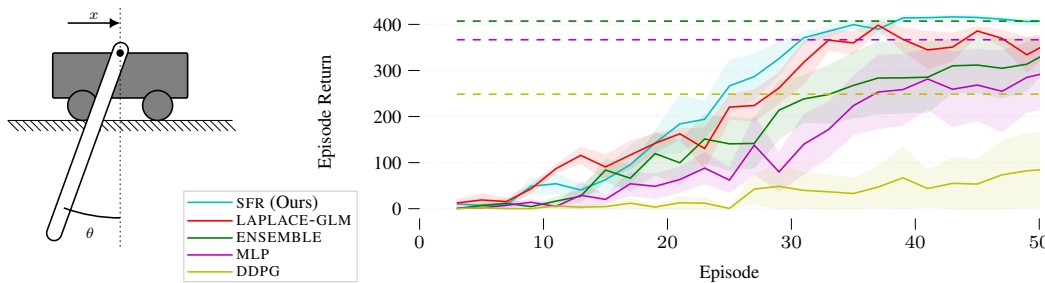

Figure A7: **Cartpole swingup with sparse reward:** Training curves show that SFR's uncertainty estimates improve sample efficiency in RL. Our method (——) converges in fewer environment steps than the baselines. The dashed lines mark the asymptotic return for the methods not coverged in the plot.

with the lowest validation loss was used as the NN MAP. Each experiment was ran for 5 seeds. The Gaussian likelihood's noise variance was set to $\sigma^2_{\text{noise}} = 1$ and trained alongside the NN's parameters.

**Data split** Each data set was split $50\% : 50\%$ into an in-distribution (ID) set $\mathcal{D}_1$ and an out-of-distribution (OOD) set $\mathcal{D}_{\text{OOD}}$ by ordering the data along the input dimension with the most unique values and splitting down the middle. The in-distribution (ID) data set $\mathcal{D}_1$ was then split $70\%$ train and $30\%$ validation. The out-of-distribution (OOD) data set $\mathcal{D}_{\text{OOD}}$ was then split $70\%$ for updates $\mathcal{D}_2$ and $30\%$ for calculating test metrics.

The results in Table 3 were obtained with the following procedure. We first trained the MLP on $\mathcal{D}_1$ and calculated SFR's sparse dual parameters. Table 3 reports SFR's test NLPD as well as the time to train the NN and calculate SFR's sparse dual parameters (Train w. $\mathcal{D}_1$). We then took the trained NN and incorproated the new data $\mathcal{D}_2$ using dual conditioning from Eq. (17) (Updates w. $\mathcal{D}_2$) . Finally, we compare incorporating new data via SFR's dual conditioning to retraining from scratch. That is, reinitializing the NN and training on $\mathcal{D}_1 \cup \mathcal{D}_2$ (Retrain w. $\mathcal{D}_1 \cup \mathcal{D}_2$).

### D.4 REINFORCEMENT LEARNING EXPERIMENT DETAILS

This section details how we configured and ran our reinforcement learning experiments.

**Environment** We consider the cartpole swingup task in MuJoCo (Todorov et al., 2012). However, we make exploration difficult by implementing a sparse reward function which returns 0 unless the reward is over a threshold value. That is, our reward function is given by,

$$\hat{r}(\boldsymbol{s}_t, \boldsymbol{a}_t) = \begin{cases} r(\boldsymbol{s}_t, \boldsymbol{a}_t), & \text{if } r(\boldsymbol{s}_t, \boldsymbol{a}_t) \geq 0.6 \\ 0, & \text{otherwise} \end{cases}$$

In all experiments we collected an initial data set using a random policy for one episode and we set action repeat as two.

**Dynamics model** In all experiments we used an MLP dynamics model with a single hidden layer of width 64 and $\tanh$ activation functions. At each episode we used Adam (Kingma & Ba, 2015) to optimize the NN parameters for 5000 iterations with a learning rate of 0.001 and a batch size of 64. We reset the optimizer after each episode. As we are performing regression we instantiate the loss function in Eq. (1) as the well-known mean squared error. This corresponds to a Gaussian likelihood with unit variance. We then set the prior precision as $\delta = 0.0001$.

**Value function learning** We use DDPG to learn the action value function. DDPG learns both a policy and a value function but we do not use the policy. In our experiments, we parameterized both the actor and critic as MLPs with two hidden layers of width 128 with ELU activations. We train them using Adam for 500 iterations at each episode, using a learning rate 0.0003 and a batch size of 512. DDPG uses a target value function to stabilize learning and for the soft target updates we used $\tau = 0.005$. DDPG is an off-policy algorithm where the exploration policy samples from a noise process, which here was a Gaussian distribution with $\sigma = 0.1$ and clipping at 0.3.

**Model predictive path integral (MPPI)** In all model-based RL experiments, we used a $H = 5$ step horizon and a discount factor of $\gamma = 0.9$. We sampled $N = 256$ trajectories for $J = 5$ iterations with $K = 32$. We used a temperature of $0.5$ and momentum $0.1$.

**MLP** As a baseline, we ran experiments using only the MLP dynamics model with no uncertainty. As can be seen in Fig. A7, this strategy is not always able to solve the task, indicated by the magenta line converging at $\sim 360$. This is expected because this strategy has no principled exploration mechanism.

**SFR** In the SFR experiments we used $M = 128$ inducing points as this seemed to be sufficient. The results in Fig. A7 indicate that SFR's uncertainty estimates are of high quality. This is because the agent is able to solve the task with less environment interaction, indicated by the cyan training curve converging quicker.

**Laplace-GGN** In the Laplace-GGN experiments we used the Laplace library from (Daxberger et al., 2021) with the full hessian structure over all of the network's weights. We then used the GLM predictions. As expected, the Laplace-GGN is also able to solve the sparse reward environment.

**Ensemble** The ensemble experiments used $5$ members where each member was an MLP with one hidden layer of width $128$ and $\tanh$ activations, *i.e.* the same architecture as the other experiments. At each episode, the ensemble was trained for $10000$ iterations using the Adam optimiser. We used a learning rate $0.0001$ and a batch size of $64$. At each time step in the planning horizon we used the following procedure to sample from the ensemble. First, the mean and variance of the ensemble members predictions were calculated. Next, we sampled from the resulting Gaussian. Fig. A7 shows that the ensemble required more environment interaction to solve the task. This is indicated by the green line not converging within the $50$ episodes shown in the figure.

**DDPG** The DDPG experiment serves as our model-free RL baseline. In the DDPG experiment we use a standard deviation of $0.6$ for the actor to help it explore the environment. We increased the width of the MLPs to $512$ and we increased the discount factor to $\gamma = 0.99$. As expected, the results show that DDPG requires a lot more environment interaction to solve the task. Fig. A7 also shows that DDPG does not converge to the same asymptotic performance as SFR. This is due to some of the random seeds not being able to successfuly explore the environment.

**Summary** Our results motivate further experiments looking into NN uncertainty quantification for model-based RL. To the best of our knowledge, we are the first to leverage the Laplace-GGN for model-based RL. Our results indicate that both the Laplace approximation and SFR could offer benefits over ensembles of NNs, which are typical in model-based RL. Although further experiments are required, our results indicate that SFR could offer better uncertainty estimates in these tasks. We believe this may be due to SFR sampling in function space as opposed to weight space like Laplace-GGN.

## E  COVARIANCE VISUALIZATION

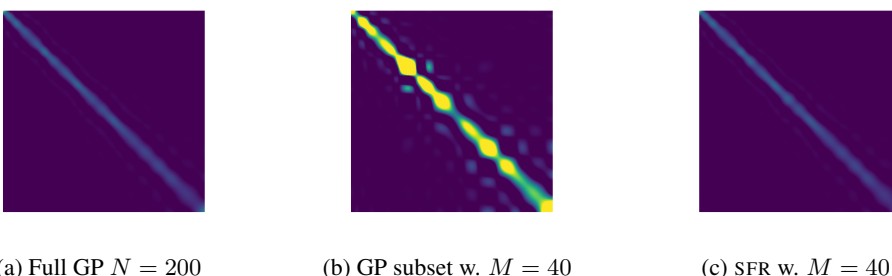

(a) Full GP $N = 200$          (b) GP subset w. $M = 40$          (c) SFR w. $M = 40$

Figure A8: **Comparison of covariance matrix for a full GP vs. GP subset vs. SFR**. The contour plot shows the covariance matrix evaluated at the training inputs $\kappa(\boldsymbol{X}, \boldsymbol{X})$ of the 1D regression example from Fig. 1.

Fig. A8 compares the covariance structure of SFR and the GP subset for the 1D regression problem in Fig. 1. It shows that SFR accurately recreates the full solution despite having a limited number of inducing points. In contrast, the GP subset (with the same computational complexity in terms of $M$) fails to accurately recreate the full solution with the same number of inducing points.

