# OpenReview forum: "Function-space Parameterization of Neural Networks for Sequential Learning"
_ICLR.cc/2024/Conference — ICLR 2024 poster_

### Official Review · Reviewer_C2Qu · 2023-10-31

**Soundness:** 3 good
**Presentation:** 3 good
**Contribution:** 3 good
**Rating:** 6
**Confidence:** 3

**Summary:**

This paper extends recent dual parameterization approaches for sparse GPs to function space inference for neural networks. This allows for efficient uncertainty estimation and the authors further formulate a continual learning objective as well as a variation that allows for incorporating new data without retraining. The experiments report competitive performance compared to Laplace- and GP-based baselines from the literature for uncertainty estimation on MNIST and CIFAR scale datasets, continual learning on MNIST and model-based reinforcement learning for the cart pole mujoco challenge.

Overall, the paper is quite rich and in terms of its technical contribution more than sufficient for a conference paper. Unfortunately, there is almost so much going on that the paper lacks in clarity and focus. The experiments tend to be somewhat small scale, so that I find it hard to argue for acceptance on a purely quantitative basis, and qualitatively the most interesting aspect - the possibility of incorporating new data without re-training - is mostly an afterthought in the empirical evaluation.

All things considered, I would lean towards rejection at this point, but with a thorough update of the manuscript would hope to increase my score over the course of the rebuttal.

EDIT: in light of the rebuttal, I would now tend towards acceptance.

**Strengths:**

- there is quite a lot going on technically, extending recent dual parameterization methods for GPs to neural networks not only for uncertainty estimation, but also continual learning
- there is a wide range of experiments, I would not have expected the model-based RL benchmark on top of the uncertainty estimation/continual learning problems
- prior work that the paper builds on is clearly referenced
- the possibility of incorporating new data without re-training is potentially practically impactful

**Weaknesses:**

- Largely due to the breadth of material covered, I find that the paper somewhat lacks depth and details in the main text. I appreciate that the authors don’t attempt to break their work into the smallest publishable units, but with the conference format being what it is, in my opinion there is also a case to be made for splitting work e.g. into a generic uncertainty estimation method and then extending it to continual learning in a separate paper in order to leave enough space for covering enough details on the method/design choices (Q1&2) and discussing the relationship with prior approaches (Q3-5).
- The title seems to place the emphasis on the continual learning part of the paper, however the empirical evaluation here is extremely small scale (nothing beyond MNIST-scale). I would find it difficult to argue for the method on a quantitative basis and would find e.g. Split CIFAR results helpful.
- Similarly the in-distribution uncertainty evaluation is a bit limited. I appreciate that these experiments were run as sanity checks, but to me it would be sufficient to leave them in the appendix (or run the full suite of uncertainty evaluation experiments on OOD/shifted data to make a more convincing quantitative case for the method).
- The evaluation of the dual parameter updates is mostly an afterthought, Table 4 seems to only show that it is possible in principle. I think this is a real shame, I’m not aware of any other near-zero cost approach for incorporating new data. I wish this was a much bigger focus in the empirical evaluation and there was more of an attempt to push the limits here, e.g. by using smaller and smaller initial datasets for learning the NN parameters, longer streams of data or two distinct datasets, e.g. MNIST/FashionMNIST or CIFAR10/SVHN where the learned parameters may be less relevant for the second datasets (essentially I would like to see the boundaries being pushed and get a sense of where the dual parameter updates start to fail).
- The continual objective seems to scale linearly in the number of tasks (both in terms of memory and computational complexity), in contrast to weight space methods that tend to add a constant overhead.

**Questions:**

1. How are the inducing points chosen? This isn’t discussed in the main text, they are simply assumed to exist around Eqs 12 & 13. The pseudocode in the appendix seems to suggest they are sampled from the real data?
2. Similarly, the continual objective in Eq 14 appears out of nowhere. I appreciate that there is a derivation in the appendix, but there is no reference to this in the main text. Does the objective correspond to some principled approximation to the posterior as in e.g. VCL or online Laplace (although these of course accumulate approximation errors)? Or is it some ad-hoc expression to maintain close predictions on old data with previous parameters?
3. The paper states at the end of the related work section “In contrast to the method presented in this paper, these functional regularization methods lack a principled way to incorporate information from all data. Instead, they rely on subset approximations”, but this seems to be the case for the present paper as well (assuming I'm not mistaken in Q1)? I understand that the dual parameters ultimately are determined as a sum over all data, yet the projection onto the inducing points does seem like a subset approximation to me, in particular since the regularization term is a sum over the inducing points, i.e. a subset.
4. Would optimizing the inducing points render the method equivalent to (Ortega et al., 2023)? If so, that seems like a highly relevant baseline/ablation.
5. The objective in Eq 14 seems quite closely related to that formulated in (Rudner et al., 2022) — which is admittedly rather generic. Would your approach fit in as a variational method or is it lacking the penalty terms related to the precision matrix?

Minor:
- I think there is a mixup in the bibtex entry for the VCL citation, this should presumably be (Nguyen et al., Variational Continual learning, ICLR 2018)
- Table 1: MAP seems to perform by far the best on half the datasets, do you have any comments on this? It might also be worth adding a HMC baseline (at least if you think of your method as a BNN method, I don’t think this is stated anywhere explicitly).
- Table 2: accuracy values are decimals rather than percent as indicated in the header
- Table 2: including the Laplace results without tuning the prior precision seems fairly pointless, it is well known across the literature that sampling from the Laplace approximation in weight space does not work well without cross-validation or online tuning. Why not use the latter if rejecting the former due to breaking the MAP property of the parameters?
- NLPD should be defined
- Laplace Kron should be referenced/cited in the experiments
- "Hessian is often infeasible and in practice it is common to adopt the GGN approximation" This seems like a slightly inaccurate statement, infeasible is calculating the full matrix, the GGN approximation is to ensure positive definiteness
- Why the two separate for loops in Algorithm 1? It seems like the SLICE variables just add notational clutter and the updates to the sum could be done in a single loop?
- The appendix mentions using double precision for the post-training computations, is that a precautionary choice or do e.g. the matrix inversions fail in single precision?

---

> ### Author Response · Authors · 2023-11-17
> **Response to Reviewer C2Qu's weaknesses**
>
> We would like to deeply thank the reviewer for taking the time to fully understand our submission and give valuable feedback. We are encouraged that you find our work novel. We have been able to greatly improve our manuscript in light of your comments. We will now reply to your comments one by one:
>
> **W1:** *Largely due to the breadth of material covered, I find that the paper somewhat lacks depth and details in the main text.  I appreciate that the authors don’t attempt to break their work into the smallest publishable units, but with the conference format being what it is, in my opinion there is also a case to be made for splitting work e.g. into a generic uncertainty estimation method and then extending it to continual learning in a separate paper in order to leave enough space for covering enough details on the method/design choices*
> **AW1:** We want to reiterate the focus of the paper. The current methods for scaling NN to GP methods rely on what they call subset approximations of the full GP. We propose using an inducing point approximation. We experimentally show that our method has a better representation at the same computational budget as the subset approximation and is competitive with other posthoc Bayesian neural network methods. We have now reworked our manuscript to emphasise the focus of our work and included some more thorough OOD experiments (see changes).
>
> At the same time, we point out that the inducing point approximation can also be used for functional regularisation in CL, essentially using the same method. Although this is not the paper's primary focus, the core methodology for both areas is the same: the inducing point approximation obtained through the dual variables or previously through a subset. Therefore, splitting up the work would reduce a CL paper because the methodology would not be novel. We agree that the connection to the previous and motivation for the CL needed improvement and have thus made changes to the manuscript.
>
> Similarly, with the dual updates the methodology is essentially the same. We agree they are a novel contribution and also included them in the paper to show that this is a promising direction for many applications, but are not the main focus of the paper. We intend to investigate these further in future work.
>
> **W2:** *The title seems to place the emphasis on the continual learning part of the paper.*
> **AW2:** We aimed for the title to be around the representative power of the inducing point posterior, which we see as relevant for both supervised learning (SL) and continual learning (CL). Nevertheless, we are open to suggestions.
>
> **W3:** *Similarly the in-distribution uncertainty evaluation is a bit limited... or run the full suite of uncertainty evaluation experiments on OOD/shifted data to make a more convincing quantitative case for the method.*
> **AW3:** We have now added OOD experiments on FMNIST/CIFAR-10. Please see our response to all reviewers.
>
> **W4:** *The continual objective seems to scale linearly in the number of tasks (both in terms of memory and computational complexity), in contrast to weight space methods that tend to add a constant overhead.*
> **AW4:** This is correct. Note that this is the case for every other function-space GP CL method; the cost for the improved performance appears. However, it is worth noting that in [1], they created a sparse GP CL method with a constant overhead like weight methods. In principle, a similar approach could be applied here, and it would involve moving inducing points around the input space as more data came rather than fixing the number of inducing points per task. We leave such a method to future work.
>
> **References**
> [1] Paul Edmund Chang, Prakhar Verma, S. T. John, Arno Solin, Mohammad Emtiyaz Khan (2023). Memory-based dual Gaussian processes for sequential learning. In *Proceedings of the 40th International Conference on Machine Learning*, PMLR 202:4035-4054.
> [2] Rasmussen, Carl Edward, and Christopher KI Williams (2006). *Gaussian Processes for Machine Learning*. Cambridge, MA: MIT press.

---

> ### Author Response · Authors · 2023-11-17
> **Response to Reviewer C2Qu's questions and minor comments**
>
> **Q1:** *How are the inducing points chosen? This isn’t discussed in the main text, they are simply assumed to exist around Eqs 12 & 13. The pseudocode in the appendix seems to suggest they are sampled from the real data?*
> **AQ1:** Thank you for spotting this. We have updated the text in Eq. (13) to state that we sample the inducing inputs from the training inputs. We have also added that investigating methods for selecting inducing inputs is an interesting direction for the future (in the conclusion). We have also referred to Alg. 1 (how we compute SFR's dual parameters) from Sec. 3.
>
> **Q3:** *The paper states at the end of the related work section “In contrast to the method presented in this paper, these functional regularization methods lack a principled way to incorporate information from all data. Instead, they rely on subset approximations”, but this seems to be the case for the present paper as well (assuming I'm not mistaken in Q1)? I understand that the dual parameters ultimately are determined as a sum over all data, yet the projection onto the inducing points does seem like a subset approximation to me, in particular since the regularization term is a sum over the inducing points, i.e. a subset.*
> **AQ3:** Following the GP literature, both the inducing points and the subset method are what we call "sparse approximations". However, as you said, SFR contains information from all the data whereas the subset method ignores some of the data to ensure computational feasibility. We get the term subset method from the GP literature where the subset method was employed prior to more modern inducing point methods [2]. If the reviewer disagrees with our word choice here we are open to changing it.
>
> **Q4:** *Would optimizing the inducing points render the method equivalent to (Ortega et al., 2023)? If so, that seems like a highly relevant baseline/ablation.*
> **AQ4:** Ortega et al., (2023) presented a preliminary approach for extending the sparse variational GP approximation from Titsias (2009), to the linearized Laplace approximation (LLA). However, this is only a "first draft" preprint on arXiv, there is only one experiment on a toy 1D regression data set, and to the best of our knowledge, there is no publically available code. Whilst we are eager to see if the authors can scale this to image data sets, we think that implementing their method and extending it for our experiments is a novel contribution and out of the scope of a baseline.
>
> **M1:** *Table 2: including the Laplace results without tuning the prior precision seems fairly pointless, it is well-known across the literature that sampling from the Laplace approximation in weight space does not work well without cross-validation or online tuning.*
> **AM1:** We report results both with and without prior precision tuning because we think this is a significant advantage of our approach. Whilst it is typical to tune the prior precision posthoc with a grid search, this can add significant overhead, which SFR does not require. Nevertheless, for the new OOD experiments, we tune the prior precision for all Laplace methods. We have also moved the UCI results without prior precision tuning to the appendix.
>
> **M2:** *The appendix mentions using double precision for the post-training computations, is that a precautionary choice or do e.g. the matrix inversions fail in single precision?*
> **AM2:** This is a good observation which we have now clarified in our manuscript. Please see Appendix A.4.

---

> ### Author Response · Authors · 2023-11-21
> **Finalised out-of-distribution (OOD) detection results**
>
> We thank Reviewer C2Qu for their suggestion to include OOD detection experiments. The new results highlight that our method performs well at detecting OOD data. Please see our comment on the "Response to all reviewers".
>
> Please can we ask the reviewer to consider increasing their score, if they are satisfied with our changes.

---

> > ### Comment · Reviewer_C2Qu · 2023-11-21
> >
> > Thank you for the clarifications and additional experiments (even if the OOD experiment is in itself a little bit limited; I completely understand that time is quite limited for the rebuttal, so it would be great to have perhaps the CIFAR distribution shifts in the camera-ready paper as well). I will increase my score.

---

### Official Review · Reviewer_7JfH · 2023-11-01

**Soundness:** 2 fair
**Presentation:** 1 poor
**Contribution:** 2 fair
**Rating:** 6
**Confidence:** 3

**Summary:**

The paper presents a new method to turn a pre-trained NN, trained with the MAP objective, into a GP model with a kernel based on the Neural tangent kernel. They then show its usefulness on many different tasks.

**Strengths:**

- The method, to the best of my knowledge is novel.
- The experiments cover a diverse set of problems

**Weaknesses:**

- My main issue with this paper is that it does not explain or derive any of its equations, not even in the supplementary. It is not clear how the authors arrived at eq. 7,8,10 & 11. It isn't even clear what the authors mean by dual parameterization (dual in what sense?) as it was never explained. While basic things like Laplace approximation and sparse GP are explained, the main points of this paper aren't.
- I am unclear about the issue with adding points to the standard GP. Once you have your kernel, given to you by the MAP trained NN, you can use it on any number of points you want. You can also train a standard sparse-GP approximation using the fixed kernel. The authors claim that this is an issue that their work solves, but I don't see any reason why it should be so and they do not explain it properly.
- Baselines in many of the experiments are very basic. For instance, in Table 1 you compare to Laplace approximation and using a simple subset selection sparse GP. Other Bayesian NN methods should be compared against, for example NN-GP, deep kernel learning, SWAG.
- I would also point out another relevent work on incremental learning by GPs Achituv et al "GP-Tree: A Gaussian process classifier for few-shot incremental learning"

**Questions:**

Please explain how you arrive at eq. 7-11

---

> ### Author Response · Authors · 2023-11-17
> **Repsonse to Reviewer 7JfH**
>
> We thank the reviewer for their comments. We will now detail how we have improved our manuscript in light of their comments.
>
> **W1:** *My main issue with this paper is that it does not explain or derive any of its equations... not clear how the authors arrived at eq. 7,8,10 & 11.... what the authors mean by dual parameterization (dual in what sense?).*
> **A1** We have added a section to the appendix (see App. A.1) to address this concern. In particular, we have addressed concerns regarding Eqs. (7–8,10–11) with the following:
> - Provided more background on the dual parameters,
>     - including why they are termed "dual" parameters (following Adam et al., 2021, who introduced the parameterization for SVGPs).
>     - including derivations of the dual parameters for the sparse variational GP ELBO.
> - Provided further derivations of the dual parameters for our MAP objective and how they relate to the dual parameters of the sparse variational GP ELBO.
>
> With regards to **Eq. (9)**, this is just the conversion of a linear model to its kernel form. Assume we have a Bayesian linear regression,
> $$
>  y_i = \mathbf{x}_i^\top \mathbf{w} + \epsilon,
> $$
> where we have a prior $\mathcal{N}(\mathbf{w}; 0, \frac{1}{\delta}I)$. If we define the function space output of the model as $f_i = \mathbf{x}_i^\top\mathbf{w}$, we can convert our weight space prior to function space as a GP. This is done by setting the mean function to the expectation of the linear model and the kernel to the covariance of the linear model (with respect to the prior). This results in a GP prior given by $\mathcal{N}(\mathbf{f}; \mathbf{0}, \frac{1}{\delta}\mathbf{X} \mathbf{X}^\top)$. If you replace the $\mathbf{x}_i^\top$ with the Jacobian you end up with Eq. (9) in the paper.
>
>
> **W2:** *I am unclear about the issue with adding points to the standard GP. Once you have your kernel, given to you by the MAP trained NN, you can use it on any number of points you want.*
> **A2:** The issue with adding new data with standard GP equations is that it's computationally infeasible for large data sets, which are common in deep learning. We thank the reviewer for spotting that we overlooked these details in the manuscript. Whilst we mentioned that the issue is computational complexity in Fig. 2, we have now also provided more details on the computational issues of standard GP updates in the appendix. Alongside details of the computational complexity of SFR's dual updates in App. A.3.
>
>
> **W3:** *You can also train a standard sparse-GP approximation using the fixed kernel. The authors claim that this is an issue that their work solves, but I don't see any reason why it should be so and they do not explain it properly.*
> **A3:** Let us start by highlighting that sparse GP approximations are largely underexplored in the NN to GP setting. Nevertheless, the reviewer is correct, technically one could use any sparse GP approximation. Note that we explicitly mention how any sparse approximation could be used in Section 3. We specifically opted for the dual parameterization approach for two reasons. First, it enables us to sparsify the GP without the extra optimisation loop required for sparse variational approximations. Second, the dual parameterization offers a low(ish) cost mechanism for incorporating new data.
>
>
>
>
> **W4:** *Baselines in many of the experiments are very basic. For instance, in Table 1 you compare to Laplace approximation and using a simple subset selection sparse GP. Other Bayesian NN methods should be compared against, for example NN-GP, deep kernel learning, SWAG.*
> **A4:** In light of the reviewer's comments, we have conducted extra experiments for the GP predictive from Immer et al. (2021b), which is the generalization of DNN2GP (Khan et al., 2019) to non-Gaussian likelihoods. To the best of our knowledge, this is the state-of-the-art NN to GP method. We have added results to the FMNIST/CIFAR-10 table (Table 1) and the UCI table (Table A5).
>
> **W5:** *I would also point out another relevent work on incremental learning by GPs Achituv et al "GP-Tree: A Gaussian process classifier for few-shot incremental learning"*
> **A5:** We thank the reviewer for pointing us to this work and does indeed seem relevant and added it to the related work section.
>
> **References**
> [1] Alexander Immer, Maciej Korzepa, and Matthias Bauer (2021b). Improving predictions of Bayesian neural nets via local linearization. In *Proceedings of The 24th International Conference on Artificial Intelligence and Statistics (AISTATS)*, volume 130 of Proceedings of Machine Learning Research, pp. 703–711. PMLR.
> [2] Mohammad Emtiyaz Khan, Alexander Immer, Ehsan Abedi, and Maciej Korzepa (2019). Approximate inference turns deep networks into Gaussian processes. In *Advances in Neural Information Processing Systems 32 (NeurIPS)*, pp. 3094–3104. Curran Associates, Inc.

---

> > ### Comment · Reviewer_7JfH · 2023-11-18
> > **Response**
> >
> > Thank you for addressing my concerns and giving much more detail on the derivation, as well as a stronger baseline comparison. I raise my score following your changes.
> >
> > I would advise further edit the paper as I feel that most of the relevant information is in the supplementary while the paper uses a lot of its space for non-critical background. It would be much better if the method was explained better, even informally, in the main text and the supplementary for all the technical details.

---

> > > ### Author Response · Authors · 2023-11-18
> > >
> > > Thank you for responding quickly and taking the time to look at our changes and update your score accordingly.
> > >
> > > We appreciate your advice regarding moving non-critical background to the appendix in favour of explaining our method in more detail. Once we receive comments from the other reviewers we will improve our manuscript further.

---

### Official Review · Reviewer_V1B4 · 2023-11-02

**Soundness:** 4 excellent
**Presentation:** 3 good
**Contribution:** 3 good
**Rating:** 8
**Confidence:** 2

**Summary:**

This paper introduces a new method for approximating a weight-space neural network using a function-space Gaussian process. To overcome the inhibitive computational complexity it then uses two techniques: a dual parameterization using a Laplacian approximation (equation 10) and sparsification (equations 12 and 13). The resulting sparse GP doesn't require any further training, but makes it easy to incorporate new data, which makes it suitable for continual learning. The paper proceeds with a series of experiments which show that the proposed model is effective at sparsifying (figure 3 and table 2), uncertainty quantification (table 1), continual learning (table 3), and incorporating new data (table 4).

**Strengths:**

I must preface this review by saying that this paper is not in my field of expertise. However, as someone who is unfamiliar with the topic, I found this paper well written and clear. The authors do a good job reviewing the set of results from the literature that they build on, before presenting their own model. They argue convincingly show that their model has practical advantages (i.e., for incorporating new data and continual learning), and the experiments seem to support the theory.

**Weaknesses:**

Perhaps the text would benefit from some pseudo-code or explanations that relate the theory to the actual practical algorithm, particularly for a reader like myself who is unfamiliar with the related literature.

I am also not too familiar with the continual learning or reinforcement learning literature, so I find it difficult to judge whether table 3 and figure 4 present a competitive set of baselines, but the results seem compelling.

Minor:

* "adpot" should be "adopt" in section 3
* "subest" shoudl be "subset" in "Sparsification via dual parameters"

**Questions:**

These might be obvious to someone more familiar with the literature, but questions that I was left with were:

* The final model (equation 12) seems to rely on inversing $\\bf{K_{zz}}$. How are the inducing points to form this matrix chosen?
* In the paragraph "GPs" it says that $\\bf{\Lambda}$ can be considered per-input noise. How so?

---

> ### Author Response · Authors · 2023-11-17
> **Response to Reviewer V1B4**
>
> We thank the reviewer for their comments and appreciate that they thought the paper was novel, well-written and clear. We will now answer the comments one by one.
>
> **W1:** *Perhaps the text would benefit from some pseudo-code or explanations that relate the theory to the actual practical algorithm, particularly for a reader like myself who is unfamiliar with the related literature.*
> **A1:** We agree that the paper does require a grasp of the literature. To help mitigate this, we have reworked the algorithm in the appendix (see A.5) and added more text to the CL section and links to dual parameters (see main paper).
>
> **W2:** *I find it difficult to judge whether table 3 and figure 4 present a competitive set of baselines.*
> **A2:** We have added further experiments to test the out-of-distribution performance of our method as well as added a new baseline (see Figure 4, Figure A7 and Table 1).
>
> **Q3:** *How are the inducing points to form this matrix chosen?*
> **A3:** We apologise for not stating this in the paper but they are currently randomly selected from the training data. We leave it for future work to pick more sophisticated sampling techniques and have mentioned both of these points in the paper.
>
> **Q4:** *In the paragraph "GPs" it says that  can be considered per-input noise. How so?*
> **A4:** The $\mathbb{\Lambda}$ is a diagnol matrix where each element is $\nabla_{f_if_i}\log p(y_i \mid f_i)$. If the likelihood is Gaussian, the derivative becomes $\frac{1}{\sigma^2}$, so the inverse of the Gaussian likelihood variance is shared across all data points. Suppose the likelihood is non-Gaussian, and we perform approximate inference. Each matrix element has a separate second derivative for each data point corresponding to the approximate normal's inverse variance. We now have a heteroskedastic Gaussian likelihood; this was also discussed further in [1] in the context of state space models.
>
> **References**
> [1] Paul E. Chang et al. (2020). Fast variational learning in state-space Gaussian process models. In *30th International Workshop on Machine Learning for Signal Processing (MLSP)*. IEEE.

---

### Author Response · Authors · 2023-11-17
**Response to all reviewers**

We are glad that all reviewers see the novelty in our work and we thank them for their valuable feedback. We have improved the manuscript in light of the reviewers' comments. The main changes are as follows (highlighted in orange in the paper):

- **Out-of-distribution (OOD) experiments (C2Qu)** We performed additional experiments to highlight the supervised learning (SL) aspect of our paper. We have performed OOD experiments on SFR trained on FMNIST and CIFAR-10. Following related literature, we report histograms showing the predictive entropy on in-distribution (ID) and OOD data. See Figs. 4 and A7. We also report the area under the receiver operating characteristic curve (AUROC), as it is a commonly used threshold-free evaluation metric for OOD detection. We have added initial AUROC scores to Table 1. Please read the paragraph titled *Out-of-distribution detection* for detailed information about the experiment. Following reviewers' recommendations (and to make space for the OOD results in the main paper) we have moved the UCI table to the appendix (now Table A5) as well as the RL figure (now Fig. A5).
- **Methodology explanation (7JfH)** To improve our method's clarity, we have added a new appendix section App. A.1 called *Dual Parameterization: Background and Derivation*. This section aims to help readers unfamiliar with some of the related work to help explain specific derivations.
- **Extra baseline (7JfH)** We have conducted extra experiments for the GP predictive from Immer et al. (2021b), which is the generalization of DNN2GP (Khan et al., 2019) to non-Gaussian likelihoods. To the best of our knowledge, this is the state-of-the-art NN to GP method. We have added results to the FMNIST/CIFAR-10 table (Table 1) and the UCI table (Table A5). We will add FMNIST/CIFAR-10 results for more seeds in the coming days.
- **Motivation for the CL regularizer (C2Qu)** We have added further details in the main text and the appendix around the motivation and derivation of our CL regularizer term and how it differs from previous approaches.
- **Small updates.** We have also made numerous updates for typos, improving clarity, adding citations, etc.


## References
* Alexander Immer, Maciej Korzepa, and Matthias Bauer (2021b). Improving predictions of Bayesian neural nets via local linearization. In *Proceedings of The 24th International Conference on Artificial Intelligence and Statistics (AISTATS)*, volume 130 of Proceedings of Machine Learning Research, pp. 703–711. PMLR.
* Mohammad Emtiyaz Khan, Alexander Immer, Ehsan Abedi, and Maciej Korzepa (2019). Approximate inference turns deep networks into Gaussian processes. In *Advances in Neural Information Processing Systems 32 (NeurIPS)*, pp. 3094–3104. Curran Associates, Inc.

---

> ### Author Response · Authors · 2023-11-20
> **Included more seeds for new experiments**
>
> As promised in our general response, we have now included the out-of-distribution (OOD) results with mean $\pm$ std over 5 seeds (for the AUROC metric). See Tables 1 and A6.
> - The OOD results are a very nice addition to our paper as it demonstrates that OOD detection is one of our methods strengths.

---

### Meta-Review · Area_Chair_aYoo · 2023-12-10

**Metareview:**

This paper develops a procedure for sequential learning by constructing a dual parameterization of the function expressed by a neural network at the trained weights, sparsifying this parameterization and updating using a new datum. This procedure is used for problems such as continual learning and out of distribution detection. The paper discusses a number of experiments on datasets from the Gaussian processes literature. The technical novelty of this paper is somewhat marginal and these ideas could be fleshed out with much greater technical sophistication.

**Justification For Why Not Higher Score:**

The reviewers have raised some legitimate concerns, e.g., weak baselines, small-scale experiments, some of which were addressed by the authors. The technical novelty of this paper is marginal.

**Justification For Why Not Lower Score:**

This paper is the natural next step in using priors on the function space parameterized by networks which can be powerful but difficult to work with. The idea explored here is quite natural, e.g., calculating the dual parameterization of the learned function using a Taylor series expansion at the fitted weights. But this simple idea could be applied in many other settings of relevance to the community.

---

### Decision · Program_Chairs · 2024-01-16

Accept (poster)